

# Toward a general calibration of the Swiss plate geophone system for
# fractional bedload transport
Tobias Nicollier[1,2], Gilles Antoniazza[3,1], Lorenz Ammann[1], Dieter Rickenmann[1], James W. Kirchner[1,2,4]
[1]Swiss Federal Research Institute WSL, Birmensdorf, 8903, Switzerland
[2]Deptartment of Environmental System Sciences, ETH Zürich, Zürich, 8092, Switzerland
[3]Institute of Earth Surface Dynamics (IDYST), University of Lausanne, Lausanne, 1015, Switzerland
[4]Deptartment of Earth and Planetary Science, University of California, Berkeley, 94720, USA
*Correspondence to*: Tobias Nicollier, Swiss Federal Research Institute (WSL), Mountain Hydrology and Mass Movements,
8903 Birmensdorf, Switzerland. E-mail: tobias.nicollier@wsl.ch. Phone: +41 77 437 35 77
**Abstract.** Substantial uncertainties in bedload transport predictions in steep streams have triggered intensive efforts to
develop surrogate monitoring technologies. One such system, the Swiss plate geophone (SPG), has been deployed and
calibrated in numerous steep water courses, mainly in the Alps. Calibration relationships linking the signal recorded by the
SPG system to the transported bedload can vary substantially between different monitoring stations, likely due to site-
specific factors such as the flow velocity and the bed roughness. Furthermore, recent controlled experiments have shown that
site-specific calibration relationships can be biased by elastic waves resulting from impacts occurring outside the plate
boundaries. Motivated by these findings, here we present a hybrid calibration procedure derived from flume experiments and
an extensive dataset of 308 calibration measurements from four different field monitoring stations. Our main goal is to
investigate the feasibility of a general, site-independent calibration procedure for inferring fractional bedload transport from
the SPG signal. First, we use flume experiments to show that sediment size classes can be distinguished more accurately
using a combination of vibrational frequency and amplitude information than by using amplitude information alone. Second,
we apply this amplitude-frequency method to field measurements to derive general calibration coefficients for ten different
grain-size fractions. The amplitude-frequency method results in more homogeneous signal responses across all sites and
significantly improves the accuracy of fractional sediment flux and grain-size estimates. We attribute the remaining site-to-
site discrepancies to large differences in flow velocity, and discuss further factors that may influence the accuracy of these
bedload estimates.
## 1   Introduction
Flood events across Europe in the summer of 2021 have illustrated the threat of flood-related hazards like bedload transport
to human life and infrastructure, especially in small and steep mountainous catchments (Badoux et al., 2014; Blöschl et al.,
2020). Understanding sediment transport processes is also essential for efforts to return rivers to their near-natural state by
restoring their continuity and re-establishing balanced sediment budgets (e.g. Brouwer and Sheremet, 2017; Pauli et al.,
2018; Logar et al., 2019; Rachelly et al., 2021). However, monitoring and predicting bedload transport still represents a
considerable challenge because of its large spatio-temporal variability (e.g. Mühlhofer, 1933; Einstein, 1937; Reid et al.,
1985; Rickenmann, 2018; Ancey, 2020). This is especially true for steep streams, because they are poorly described by
traditional bedload transport equations, which have mainly been developed for lower-gradient channels (e.g. Schneider et al.,





2016). Predicting sediment transport in steep channels is challenging, notably due to the presence of macro-roughness
elements affecting both the flow resistance and the flow energy (e.g. Manga and Kirchner, 2000; Yager et al., 2007, 2012;
Bathurst, 2007; Nitsche et al., 2011; Rickenmann and Recking, 2011; Prancevic and Lamb, 2015). It is further complicated
by a sediment supply that varies in both space and time, due in part to cycles of building and breaking of an armoring layer
at the riverbed (e.g. Church et al., 1998; Dhont and Ancey, 2018; Rickenmann, 2020; Piantini et al., 2021).

Bedload transport equations established for lower-gradient streams typically result in substantial errors when applied to
steep streams, motivating the development of new indirect monitoring techniques for steep mountain channels (e.g. Gray et
al., 2010; Rickenmann, 2017). Indirect monitoring techniques provide large spatial coverage of river transects at high
temporal resolution, reduce personal risk related to in-stream sampling, and enable consistent data collection at widely
varying flow conditions including floods (e.g. Gray et al., 2010; Rickenmann, 2017; Geay et al., 2020; Bakker et al., 2020;
Choi et al., 2020; Le Guern et al., 2021). The drawback of these monitoring technologies is that in order to provide
quantitative measurements, they require intensive calibration through direct bedload sampling with retention basins
(Rickenmann and McArdell, 2008), slot samplers (e.g. Habersack et al., 2017; Halfi et al., 2020) or mobile bag samplers
(e.g. Bunte et al., 2004; Dell'Agnese et al., 2014; Hilldale et al., 2015; Mao et al., 2016; Kreisler et al., 2017; Nicollier et al.,
2021a).

Among indirect monitoring techniques, the Swiss plate geophone (SPG) system has been deployed and tested in more
than 20 steep gravel-bed streams and rivers, mostly in the European Alps (Rickenmann, 2017). Typically, linear or power-
law calibration relationships have been developed between measured signal properties and bedload transport characteristics
(Rickenmann et al., 2014; Wyss et al. 2016a; Kreisler et al., 2017; Kuhnle et al., 2017). Such calibration equations facilitate
spatio-temporal estimates of bedload fluxes, absolute estimates of bedload fluxes and bedload grain-size distributions, and
the detection of the start and end of bedload transport. However, these equations have required calibration against
independent bedload transport measurements from each individual field site, because until now we have lacked generally
applicable signal-to-bedload calibration equations that are valid in multiple field settings. Although the similarities between
calibration relationships at various field sites are encouraging, it is not well understood why the linear calibration coefficients
for total mass flux can vary by about a factor of 20 among individual samples from different sites, or by about a factor of six
among the mean values from different sites (Rickenmann et al., 2014; Rickenmann and Fritschi, 2017). Given the substantial
field effort required for calibration campaigns, a generally applicable calibration equation would represent a significant
advance.

Numerous studies have reported successful calibration of impact plate systems in laboratory flumes (e.g. Bogen and
Møen, 2003; Krein et al., 2008; Tsakiris et al., 2014; Mao et al., 2016; Wyss et al., 2016b,c; Kuhnle et al., 2017; Chen et al.,
2021), although transferring these flume-based calibrations to the field remains challenging. Nonetheless, controlled flume
experiments are valuable because they allow us to systematically explore relationships between the recorded signal, the
transport rates of different sediment size fractions, and the hydraulic conditions. For example, the experiments of Wyss et al.
(2016b) showed that higher flow velocities induce a weaker SPG signal response per unit of transported sediment. More
recent controlled experiments have highlighted another important site-dependent factor influencing the SPG signal response,
namely the grain-size distribution (GSD) of the transported bedload (Nicollier et al., 2021a), where coarser grain mixtures
were shown to yield a stronger signal response per unit bedload weight.

Subsequent impact tests and flume experiments showed that this grain-size dependence arises because the impacts
plates are insufficiently isolated from their surroundings (Antoniazza et al., 2020; Nicollier et al., 2021b). The elastic wave
generated by an impact on or near a plate was found to propagate over several plate lengths, contaminating the signals
recorded by neighboring sensors within a multiple plate array. Nicollier et al. (2021b) introduced the notion of "apparent
packets" to define the portions of the recorded signal that were generated by such extraneous particle impacts.



The main goal of this contribution is to examine the feasibility of a general, site-independent signal conversion
procedure for fractional bedload flux estimates. We follow a comprehensive hybrid signal conversion approach that
encompasses a set of full-scale controlled flume experiments conducted at an outdoor flume facility, as well as 308 field
calibration measurements performed with direct sampling methods at four different bedload monitoring stations in
Switzerland between 2009 and 2020. We present the amplitude-frequency (AF) method, aiming to reduce the bias introduced
by apparent packets in the relationship between the signal characteristics and the particle size. Finally, we compare the
performance of this novel AF method against the purely amplitude-histogram (AH) method developed by Wyss et al.
(2016a) for fractional and total bedload flux estimates as well as for characteristic grain-size estimates.

## 2    Methods

### 2.1    The SPG system

The Swiss plate geophone (SPG) consists of a geophone sensor fixed under a steel plate of standard dimensions 492 mm x
358 mm x 15 mm (Fig. 1a; Rickenmann, 2017). The geophone (GS-20DX by Geospace technologies; www.geospace.com)
uses a magnet moving inside an inertial coil (floating on springs) as an inductive element. The voltage induced by the
moving magnet is directly proportional to its vertical velocity resulting from particle impacts on the plate. Typically, a SPG
array includes several plates next to each other, acoustically isolated by elastomer elements and covering the river cross-
section. The array is either embedded in a concrete sill or fixed at the downstream face of a check dam. A detailed
description of the SPG system can be found in Rickenmann et al. (2014). Due to data storage limitations, field stations
usually do not continuously record the full raw 10 kHz geophone signal. Instead, it is typically preprocessed, and summary
values, such as the maximum amplitude and the number of impulses, are recorded at one-minute intervals. However, for the
relatively short duration of a single calibration measurement, ranging from a few seconds to one hour, the full raw signal is
recorded (Fig. 1b).

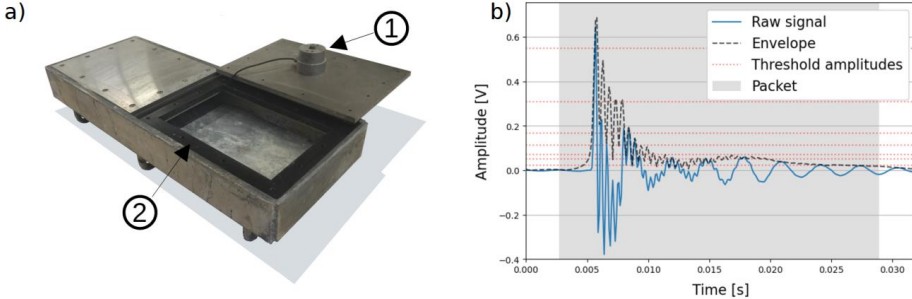


**Figure 1: (a) Swiss plate geophone (SPG) system before installation (see Fig. 3). Each plate is equipped with a uniaxial geophone**
**sensor fixed in a watertight aluminum box (1) attached to the underside of the plate. The plates are acoustically isolated from each**
**other by elastomer elements (2). (b) Example of a packet (grey area) detected by the SPG system. A packet begins 20 time steps**
**(i.e., 2 ms) before the signal envelope crosses the lowest amplitude threshold of 0.0216 V and ends 20 time steps after the last**
**crossing of the lowest amplitude threshold (see Sect. 2.4).**





### 2.2 Field calibration measurements

To test the AF and AH methods, this study uses 308 calibration measurements from four Swiss bedload monitoring stations equipped with SPG systems (Fig. 2; Table 1). Field calibration samples were collected at the Albula, Navisence and Avançon de Nant stations, and extensive calibration efforts have been undertaken at the fourth field station, Erlenbach, since 2009 (Rickenmann et al., 2012). The Erlenbach offers an interesting comparison with the other sites due to different channel and flow characteristics upstream of the SPG plates. Field calibrations at each of the four sites consist of the following steps: (i) direct bedload sampling downstream of an impact plate using either crane-mounted net samplers adapted from Bunte traps (Bunte et al., 2004; Dell'Agnese et al., 2014; Nicollier et al., 2019; Fig. 2a, b), automated basket samplers (Rickenmann et al., 2012; Fig. 2d) or manual basket samplers (Fig. 2c), (ii) synchronous recording of the raw geophone signal, (iii) sieving and weighing of bedload samples using ten sieve classes (Table 3), and (iv) comparing the fractional bedload mass of each sample to the packet histogram data to derive the corresponding calibration coefficient $k_{b,i,j}$. A more detailed description of the sampling procedure is reported in Supporting Information S1.

a)
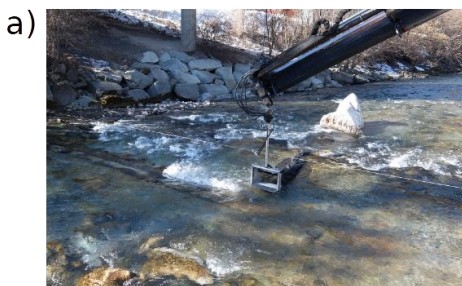

b)
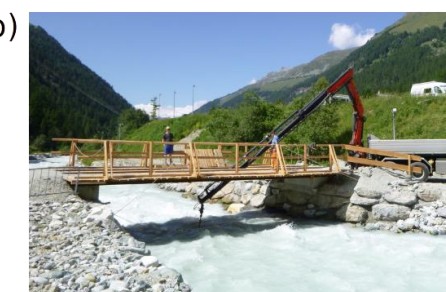

c)
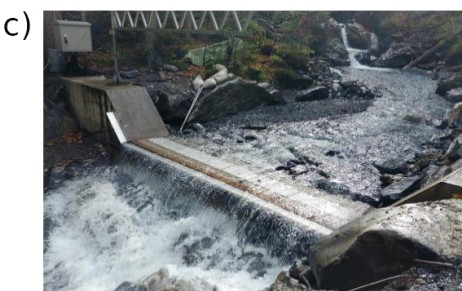

d)
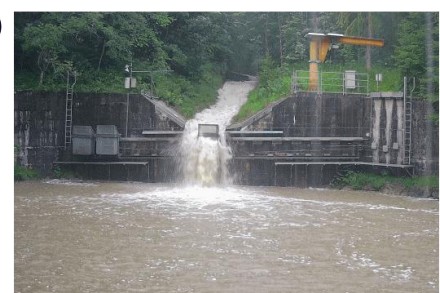

**Figure 2: The four Swiss bedload monitoring stations at which raw Swiss plate geophone signals have been recorded during calibration measurements. The stations are installed at the following streams: a) Albula, b) Navisence, c) Avançon de Nant and d) Erlenbach. Pictures a) and c) were taken during low-flow conditions. Pictures b) and d) show calibration measurements with the crane-mounted net sampler and the automated basket sampler, respectively, at high flows.**





**Table 1: Channel and flow characteristics based on *in situ* measurements during the calibration campaigns at the four field sites.**
**The year of the field calibration campaigns, the sampling technique and the number of collected samples are also indicated.**

| Field site | Location (canton) | Bed slope [%] [a] | Mean flow velocity $V_f$ [m s⁻¹] [b] | No. of plates | Year | Sampling technique | No. of samples |
|---|---|---|---|---|---|---|---|
| Albula [c] | Tiefencastel (Grisons) | 0.7 | 2.6 | 30 | 2018 | crane-mounted net sampler | 51 |
| Navisence [c] | Zinal (Valais) | 3 | 3.2 | 12 | 2019 | crane-mounted net sampler | 80 |
| Avançon de Nant [d] | Les Plans-sur-Bex (Vaud) | 4 | 1.3 | 10 | 2019/2020 | manual basket sampler | 55 |
| Erlenbach [e] | Alpthal (Schwyz) | 16 | 5.0 | 2 | Since 2009 | automatic basket sampler | 122 |

[a] Gradient measured upstream of the SPG plates. At the Erlenbach, this gradient is the slope of the artificial approach flow channel
upstream of the SPG system.
[b] Depth-averaged mean flow velocities measured during the calibration measurements.
[c] More information on the sites is available in Nicollier et al. (2021a).
[d] More information on the site is available in Antoniazza et al. (2021).
[e] More information on the site is available in e.g. Rickenmann et al. (2012), Wyss et al. (2016c), Rickenmann et al. (2018).
**2.3  Controlled flume experiments**
The first part of the signal conversion procedure described in this study is based on controlled flume experiments conducted
at the outdoor flume facility of the Oskar von Miller institute of TU Munich in Obernach, Germany. At this facility, we
reconstructed the bed characteristics of the Albula, Navisence and Avançon de Nant field sites, one after another, in a flume
test reach with dimensions of 24 m x 1 m equipped with two impact plates (Fig. 3). Each site reconstruction used bedload
material collected during field calibration measurements, and we adjusted the flow velocity, flow depth, and bed roughness
to match the respective field observations. A detailed description of the original flume setup and the performed experiments
can be found in Nicollier et al. (2020). In this paper, we primarily use the single-grain-size experiments conducted in 2018
with the flume configured to match conditions at the Albula field site (Table 2). Single-grain-size experiments consisted of
feeding the flume with a fixed number of grains for each of the ten particle-size classes described in Sect. 2.2 above. While
these particles were being transported over the SPG system, the full raw geophone signal was recorded. Up to 33 repetitions
were conducted until a representative range of amplitude and frequency values for each grain-size class were obtained
(Nicollier et al., 2021a). The same procedure was repeated for two different flow velocities ($V_f$ = 1.6 m s⁻¹ and 2.4 m s⁻¹). The
obtained information was then used to derive empirical relationships between the mean particle size $D_{m,j}$ and the packet
envelope's amplitude $MaxAmp_{env}$ and the ratio $MaxAmp_{env} / f_{centroid}$, as described in Sect. 2.5.2 below.
To illustrate the AF and AH methods and their respective performance, we use flume experiments that mimic the
Avançon de Nant field site, but with the addition of a 4 m wooden partition wall (Fig. 3) that shields one geophone plate
from impacting particles (Nicollier et al., 2021b). With this modified setup, single-grain-size experiments were run using
grains from each of the 10 particle-size classes, resulting in a total of 51 runs (Table 2). The flow velocity was set to 3 m s⁻¹
to facilitate particle transport through the narrower flume section and is therefore not representative for the Avançon de Nant
site, where typical flow velocities were roughly 1.3 m s⁻¹.



Earth **Surface**
**Dynamics**
Discussions



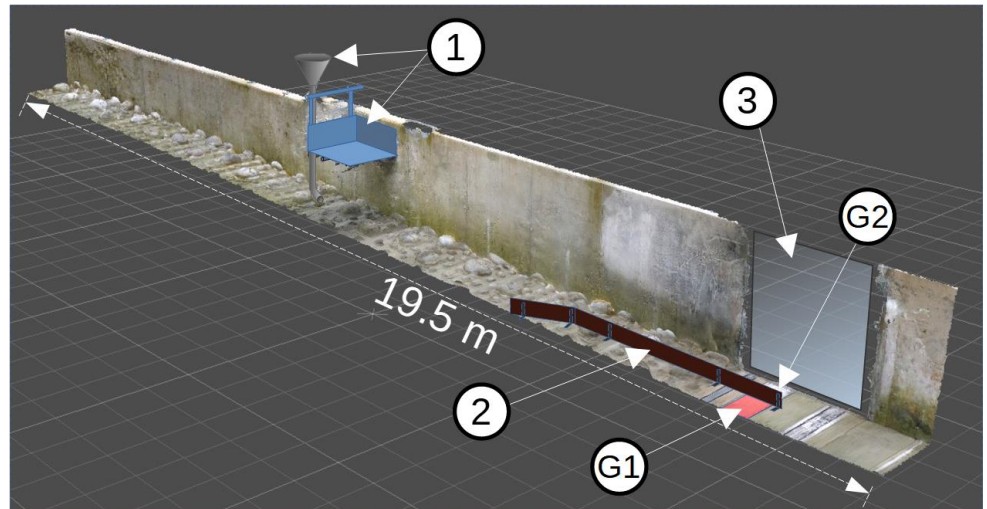


**Figure 3: Oblique view of the Obernach flume test reach with total length of 24 m and width of 1 m. The bed surface is paved with**
**particles with diameters equaling the characteristic $D_{67}$ and $D_{84}$ sizes of the natural beds of the reconstructed sites. Grains were**
**fed into the channel 8 m upstream from the SPG system location (G1 and G2) using either a vertical feed pipe or a tiltable basket**
**(1). The sensor plate G1 (in red) was shielded from direct particle impacts by the 4 m long removable partition wall (2). The**
**partition wall and the impact plates were decoupled from each other by a 2 mm vertical gap to prevent disturbances of the**
**recorded signal. Plexiglas walls (3) on each side of the flume facilitated video recordings of the experiments.**

**Table 2: Flume and hydraulic characteristics for the reconstruction of the Albula and the Avançon de Nant field sites.**

| Parameter | Units | Reconstructed field site setup | |
| --- | --- | --- | --- |
| | | Albula (without partition wall) | Avaçon de Nant (with partition wall) |
| Flume width | m | 1.02 | 1.02 |
| Flume gradient of the natural bed | % | 0.7 | 4.0 |
| Bed surface $D_{67}$ [a] | mm | 120 | 200 |
| Bed surface $D_{84}$ [a] | mm | 190 | 320 |
| Number of $D_{67}$-particles/m$^2$ | m$^{-2}$ | 15.0 | 5.0 |
| Number of $D_{84}$-particles/m$^2$ | m$^{-2}$ | 5.0 | 2.5 |
| Min. water depth above SPG | m | 0.79 | 0.35 |
| Max. water depth above SPG | m | 0.91 | 0.35 |
| Min. flow velocity 10 cm above SPG [b] | m s$^{-1}$ | 1.6 | 3.0 |
| Max. flow velocity 10 cm above SPG [b] | m s$^{-1}$ | 2.4 | 3.0 |
| Min. unit discharge | m$^2$ s$^{-1}$ | 1.6 | 0.8 |
| Max. unit discharge | m$^2$ s$^{-1}$ | 2.4 | 0.8 |
| Number of different flow velocity settings | - | 2 | 1 |
| Total number of single-grain-size experiments | - | 355 | 51 |
| Total number of tested particles | - | 10705 | 2485 |

[a] On the basis of line-by-number pebble counts at the natural site and a photo-sieving based granulometric analysis with BASEGRAIN
software (Detert and Weitbrecht, 2013).
[b] Flow velocities measured with the OTT MF Pro magnetic-inductive flow meter.





### 2.4 The amplitude-histogram method


Wyss et al. (2016a) introduced the packet-based amplitude-histogram (AH) method to derive grain-size information from
geophone signals. A packet is defined as a brief interval, typically lasting 5 to 30 milliseconds, reflecting a single impact of a
particle on a plate (Fig. 1b); it begins and ends when the signal envelope crosses a threshold amplitude of 0.0216 V. The
signal envelope is computed in Python with the Hilbert transform (Jones et al., 2002), yielding a continuous time series
reflecting the total energy in the signal. Each packet's maximum amplitude is then used to assign it to a predefined amplitude
class $j$ (Table 3), yielding a packet-based amplitude histogram (e.g. Fig. 4 in Wyss et al., 2016a). Each amplitude class $j$ is
related to a corresponding grain-size class through the following relationship between the mean amplitude $A_{m,j}$ [V] and the
mean particle size $D_{m,j}$ [mm]:
$$A_{m,j} = 4.6 \cdot 10^{-4} \cdot D_{m,j}^{1.71} . \qquad (1)$$

The coefficients in Eq. (1) were determined using 31 basket samples collected at the Erlenbach for which the maximum
geophone amplitude was analyzed as a function of the B-axis of the largest particle found in the sample (Wyss et al., 2016a).
The grain-size classes are delimited by the size of the meshes $D_{sieve,j}$ used to sieve the bedload samples from field
calibration measurements. It is assumed that the number of packets between two thresholds is related to the fractional
bedload mass between the respective sieve sizes (Wyss et al., 2016a). In the present study, we have extended the seven size
classes used by Wyss et al. (2016a) to ten classes, in order to assess the performance of the AH and AF methods for larger
particles.

**Table 3: Size classes $j$ derived from sieve mesh sizes $D_{sieve,j}$ (for classes 1 to 7) according to Wyss et al. (2016a), and mean particle**
**diameter $D_{m,j}$, amplitude-histogram thresholds $th_{ah,j}$ derived from Eq. (1), and amplitude-frequency thresholds $th_{af,low,j}$ and $th_{af,up,j}$**
**derived from Eq. (4) and (5), respectively. Particles in classes 8 to 10 were manually sorted on the basis of linearly extrapolated**
**$D_{m,j}$ values. The value of $D_{m,j}$ for the largest class (10) in brackets is an estimate, because this size class is open-ended and thus the**
**mean varied somewhat from site to site.**

| Class $j$ | $D_{sieve,j}$ | $D_{m,j}$ | $th_{ah,j}$ | $th_{af,low,j}$ | $th_{af,up,j}$ |
|---|---|---|---|---|---|
| [-] | [mm] | [mm] | [V] | [V] | [V Hz$^{-1}$] |
| 1 | 9.5 | 12.3 | 0.0216 | 0.0132 | $1.55 \cdot 10$-5 |
| 2 | 16.0 | 17.4 | 0.0527 | 0.0364 | $2.33 \cdot 10$-5 |
| 3 | 19.0 | 21.8 | 0.0707 | 0.0509 | $4.45 \cdot 10$-5 |
| 4 | 25.0 | 28.1 | 0.1130 | 0.0868 | $7.67 \cdot 10$-5 |
| 5 | 31.4 | 37.6 | 0.1670 | 0.1362 | $1.78 \cdot 10$-4 |
| 6 | 45.0 | 53.2 | 0.3088 | 0.2725 | $3.93 \cdot 10$-4 |
| 7 | 63.0 | 71.3 | 0.5489 | 0.5244 | $7.05 \cdot 10$-4 |
| 8 | 80.7 | 95.5 | 0.8378 | 0.8489 | $1.56 \cdot 10$-3 |
| 9 | 113.0 | 127.9 | 1.4919 | 1.6342 | $2.79 \cdot 10$-3 |
| 10 | 144.7 | (171.5) | 2.2760 | 2.6438 | - |


### 2.5 The amplitude-frequency method


In a recent study, Nicollier et al. (2021b) showed that the SPG system is sensitive to extraneous particle impacts despite the
isolating effect of the elastomer. Extraneous signals at individual geophone plates can arise from impacts occurring on
neighboring plates, or from impacts on the concrete sill surrounding the SPG array. The elastic waves generated by such
impacts can reach multiple geophone sensors with enough energy to be recorded as "apparent" packets. Thus, packet



histograms (i.e. counts of the number of packets per class *j*) are subject to a certain bias, especially in the lower size classes.
The degree of bias was found to depend mainly on two factors. First, coarser grain sizes of transported bedload were shown
to generate more apparent packets. Second, more apparent packets were recorded, for a given bedload mass, at transects
containing more SPG plates. Nicollier et al. (2021b) showed that packet characteristics such as the start time, the amplitude
and the frequency help in identifying apparent packets and filtering them out from the final packet histograms. This filtering
method was subsequently applied to all four field calibration datasets (Albula, Navisence, Avançon de Nant and Erlenbach)
and helped to reduce the differences between the site-specific mean calibration relationships for the total bedload flux by
about 30% (Nicollier et al., 2021b). Based on these observations, the present study proposes an amplitude-frequency (AF)
method as an adaptation of the amplitude-histogram (AH) method presented by Wyss et al. (2016a). By introducing two-
dimensional (amplitude and centroid frequency) size class thresholds, the new method aims to reduce the effect of apparent
packets and improve the accuracy of fractional bedload flux estimates.
**2.5.1   Centroid frequency**
According to the Hertz contact theory, the frequency at which a geophone plate vibrates is controlled by the size of the
colliding particle (Johnson, 1985; Thorne, 1986; Bogen and Møen, 2003; Barrière et al., 2015; Rickenmann, 2017). In the
present study, the frequency spectrum of a packet is characterized by the spectral centroid $f_{\text{centroid}}$. It represents the center of
mass of the spectrum and is computed as
$$f_{\text{centroid}} = \frac{\sum f_n \cdot A_{\text{FFT},n}}{\sum A_{\text{FFT},n}} \tag{2}$$

where $A_{\text{FFT},n}$ [V·s] is the Fourier amplitude (computed with the Fast Fourier Transform FFT) corresponding to the frequency
$f_n$ [Hz]. Following Wyss et al. (2016b), before applying the FFT, each packet is preprocessed in two steps. First, a cosine
taper is applied at the edges of a max. 8 ms time window around the peak amplitude of each packet. Second, the signal
contained in this time window is zero-padded on either side to reach an optimal number of sample points *nFFT*. The taper is
used to smooth the transition between the packet and the concatenated zeros, and to suppress spectral leakage, which results
in a more accurate amplitude spectrum. The value of *nFFT* was set to $2^7$ in order to adequately resolve the amplitude
spectrum of the raw signal contained in the max. 8 ms time window. This time window focuses on the first arrival waveform
to obtain a more accurate evaluation of the high-frequency content of the packet (Nicollier et al., 2021b). The single-sided
Fourier transform of the processed packet is then computed in order to extract $A_{\text{FFT}}$ and derive $f_{\text{centroid}}$ (Eq. 2). A decrease
in $f_{\text{centroid}}$ with increasing particle size was observed for different bedload surrogate monitoring techniques (Belleudy et al.,
2010; Uher and Benes, 2012; Barrière et al., 2015). Furthermore, $f_{\text{centroid}}$ has the advantage of showing weaker dependency
on the flow velocity and transport mode than the maximum registered packet amplitude (Wyss et al. 2016b; Chen et al.,
2021). As shown by Nicollier et al. (2021b), $f_{\text{centroid}}$ also contains information about the impact location of a packet-
triggering particle. Because high frequencies are more rapidly attenuated than low frequencies along the travel path of a
seismic wave, (apparent) packets triggered by impacts on a given plate typically have higher $f_{\text{centroid}}$ values than packets
triggered by impacts occurring beyond that plate's boundaries.
**2.5.2   Flume-based amplitude-frequency thresholds**
The transported bedload mass associated with an individual signal packet is strongly dependent on the size of the impacting
particle. Inferring sediment transport rates from SPG signals thus requires assigning each packet to a corresponding sediment
size class using threshold values of packet characteristics. Wyss et al. (2016a) derived size class thresholds (or AH
thresholds) of packet peak amplitude from field measurements (Eq. 1). In the present study, we derive size class thresholds



of packet amplitude and frequency from the single-grain-size experiments conducted at the flume facility using the Albula
setup (Nicollier et al., 2021a). For each class $j$, the lower threshold $th_{\mathrm{af,low},j}$ is based on the maximum amplitude of the
packet's envelope $MaxAmp_{\mathrm{env}}$ [V] and the upper threshold $th_{\mathrm{af,up},j}$ is based on the ratio $MaxAmp_{\mathrm{env}}/f_{\mathrm{centroid}}$ [V Hz$^{-1}$].
Compared to the raw signal, the envelope has the advantage of returning the magnitude of the analytical signal and thus
better outlines the waveform by omitting the harmonic structure of the signal. Similar combinations of amplitude and
frequency have been used to infer particle sizes and improve the detectability of bedload particles in previous studies
involving impact plates (Tsakiris et al., 2014; Barrière et al., 2015;  Wyss et al., 2016b; Koshiba and Sumi, 2018) and pipe
hydrophones (Choi et al., 2020).
The lower and upper amplitude-frequency (AF) thresholds are obtained as follows. First, all packets recorded during
the single-grain-size experiments (without the partition wall) are filtered with respect to the following criterion adapted from
Nicollier et al. (2021b):
$$Criterion:\ f_{\mathrm{centroid}} > a_{\mathrm{c}} \cdot e^{(b_{\mathrm{c}} \cdot MaxAmp_{\mathrm{env}})}\ ,\qquad(3)$$
with $a_{\mathrm{c}}$= 1980 Hz and $b_{\mathrm{c}}$= -1.58 V$^{-1}$. Packets that do not meet this criterion are considered as apparent packets and are
ignored in the further analysis. The values for the linear coefficient $a_c$ and the exponent $b_c$ were obtained through an
optimization process discussed below. The next step consists in fitting a power-law least-squares regression line through the
75$^{\mathrm{th}}$ percentile amplitude $MaxAmp_{\mathrm{env,75th},j}$ and amplitude-frequency $(MaxAmp_{\mathrm{env}}/f_{\mathrm{centroid}})_{\mathrm{75th},j}$ values of each class $j$
(Fig. 4), resulting in the following two equations:
$$MaxAmp_{\mathrm{env,75th},j} = 1.66 \cdot 10^{-4} \cdot D_{\mathrm{m},j}^{\,1.95},\ \text{ and}\qquad(4)$$
$$\left(\frac{MaxAmp_{\mathrm{env}}}{f_{\mathrm{centroid}}}\right)_{\mathrm{75th},j} = 2.26 \cdot 10^{-8} \cdot D_{\mathrm{m},j}^{\,2.36}.\qquad(5)$$
Finally, the lower and upper threshold values $th_{\mathrm{af,low},j}$ and $th_{\mathrm{af,up},j}$ are obtained by replacing $D_{\mathrm{m},j}$ in Eq. (4) and (5)
with the lower ($D_{\mathrm{sieve},j}$) and upper ($D_{\mathrm{sieve},j+1}$) sieve sizes, respectively (Table 3 and triangles in Fig. 5). The advantage in
fitting functions such as Eq. (4) and (5) is that they allow the computation of thresholds for any classification of particle
(sieve) sizes.
Particularly for the largest particles, apparent packets can greatly outnumber real packets. Due to their relatively small
amplitudes, these apparent packets can substantially dilute the average signal response associated with the largest grain sizes
(see the red boxplots in Fig. 5). However, filtering out apparent packets reveals a clear relationship, which would otherwise
be obscured, between the mean particle size $D_{\mathrm{m},j}$ and both the amplitude $MaxAmp_{\mathrm{env}}$ and the ratio $MaxAmp_{\mathrm{env}}/f_{\mathrm{centroid}}$
(see the blue boxplots in Fig. 5).





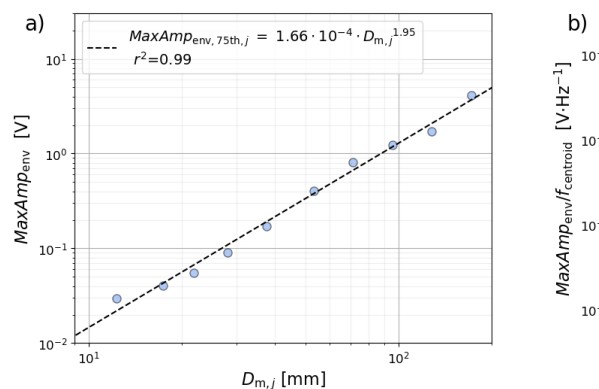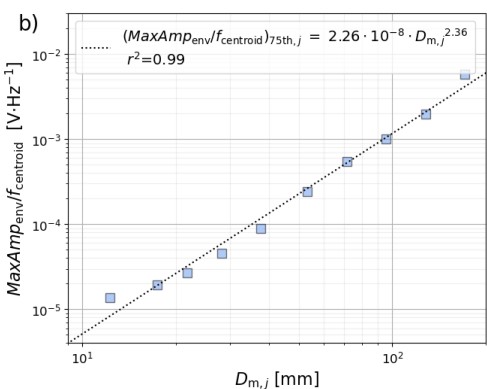

Figure 4: Power-law least-squares regression relationships between the mean particle diameter $D_{m,j}$ and the 75th percentile of the (a) amplitude $MaxAmp_{env,75th,j}$ and (b) amplitude-frequency $(MaxAmp_{env}/f_{centroid})_{75th,j}$ values obtained from the single-grain-size experiments after filtering out apparent packets using the filtering criterion in Eq. (3).

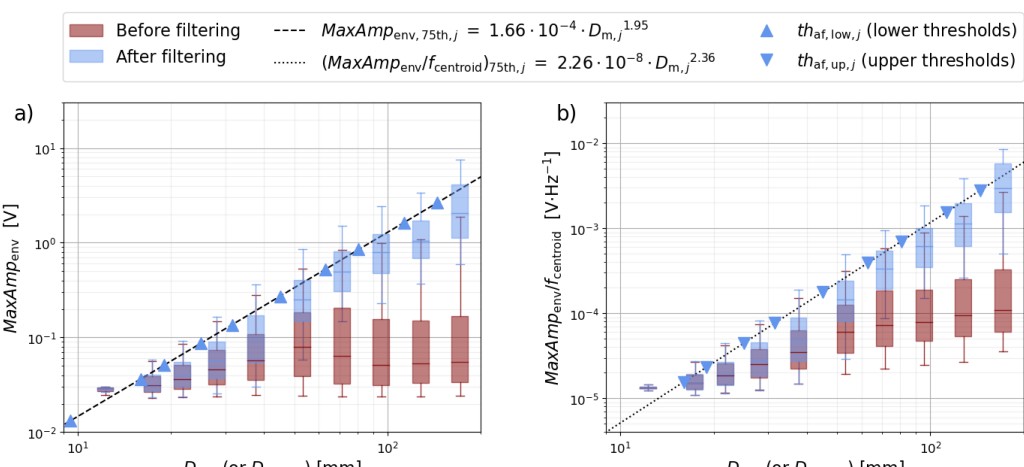

Figure 5: Range of signal responses obtained from the single-grain-size experiments before (red boxes) and after (blue boxes) filtering out apparent packets using the filtering criterion in Eq. (3), with (a) the maximum amplitude of the envelope $MaxAmp_{env}$ and (b) the ratio $MaxAmp_{env}/f_{centroid}$ as functions of the mean particle diameter $D_{m,j}$. In (a), the lower threshold values $th_{af,low,j}$ are obtained by replacing $D_{m,j}$ with the lower sieve sizes ($D_{sieve,j}$) in the equation of the dashed power-law regression line (Eq. 4). In (b), the upper threshold values $th_{af,up,j}$ are obtained by replacing $D_{m,j}$ with the upper sieve sizes ($D_{sieve,j+1}$) in the equation of the dotted power-law regression line (Eq. 5).

### 2.5.3 Application to field calibration measurements

The lower and upper thresholds $th_{af,low,j}$ and $th_{af,up,j}$ obtained from the filtered flume experiments can also be used for the field calibration datasets, if the SPG apparatus and the geophone data recording and preprocessing routines are identical in both cases. The following steps will now lead us to the final general calibration coefficients $k_{b,j,gen}$ (Fig. 6). First, for each field measurement $i$, the thresholds $th_{af,low,j}$ and $th_{af,up,j}$ are used for counting the number of packets per class $j$ from the





recorded geophone signal. Second, a sample- and class-specific calibration coefficient $k_{b,i,j}$ with units [kg$^{-1}$] is obtained by
dividing the number of recorded packets $PACK_{i,j}$ by the sampled fractional mass $M_{meas,i,j}$ as follows:

$$k_{b,i,j} = \frac{PACK_{i,j}}{M_{meas,i,j}}.$$ (6)

Finally, the general calibration coefficient $k_{b,j,gen}$ is computed for each class $j$ using

$$k_{b,j,gen} = \frac{1}{N_{stations}} \sum_{stations} k_{b,j,med,station},$$ (7)

where $k_{b,j,med,station}$ is the site-specific median calibration coefficient, and $N_{stations}$ is the number of stations. Even though
the number of calibration measurements differs from site to site, each coefficient $k_{b,j,med,station}$ in Eq. (7) is equally
weighted in order to give the same importance to site-specific factors possibly affecting the signal response at each site.

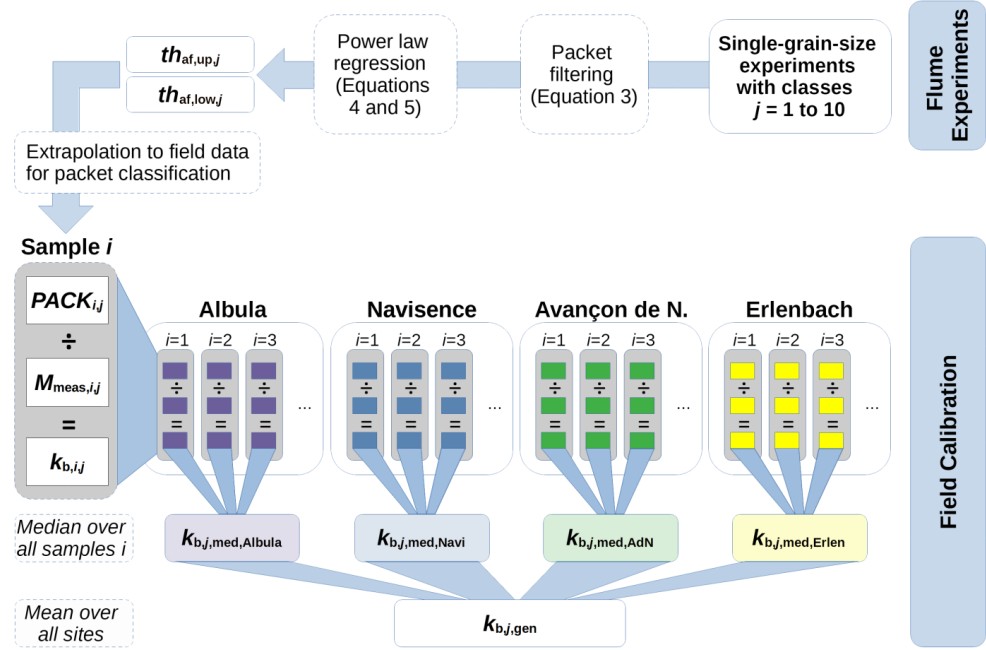


**Figure 6: Workflow leading from the single-grain-size flume experiments with particles from ten size classes $j$ (top right) to the**
**final array of general calibration coefficients $k_{b,j,gen}$. Central elements are the lower and upper threshold values $th_{af,low,j}$ and**
**$th_{af,up,j}$, the number of recorded packets $PACK_{i,j}$ per sample $i$ and class $j$, the sampled fractional mass $M_{meas,i,j}$, the sample- and**
**class- specific calibration coefficient $k_{b,i,j}$, and finally the site-specific median calibration coefficient $k_{b,j,med,station}$. To enable a**
**comparison with the AH method developed by Wyss et al. (2016a), the "Field Calibration" part of the workflow was also carried**
**out with the AH thresholds $th_{ah,j}$ (see Table 3).**
At this point, the single array of calibration coefficients $k_{b,j,gen}$ is applied as follows to each field
calibration measurement $i$ in order to obtain fractional bedload mass estimates $M_{est,i,j}$:

$$M_{est,i,j} = k_{b,j,gen} \cdot PACK_{i,j}.$$ (8)



Rickenmann and Fritschi (2017) showed that bedload mass estimates derived from SPG measurements are more accurate at
higher transport rates. The estimated fractional bedload mass $M_{\text{est},i,j}$ can be converted to a unit fractional transport rate
$q_{\text{b,est},i,j}$ [ kg m$^{-1}$ s$^{-1}$] using:

$$q_{\text{b,est},i,j} = \frac{1}{w_{\text{p}} \cdot n_{\text{p}}} \cdot \frac{M_{\text{est},i,j}}{\Delta t_i}, \tag{9}$$

where $w_{\text{p}}$ is the standard width of an impact plate (0.5 m), $n_{\text{p}}$ is the number of plates (which may include the whole transect,
or a section of particular interest), and $\Delta t_i$ is the sampling duration in seconds. Finally, the estimated unit total bedload flux
$q_{\text{b,tot,est},i}$ can be computed as follows:

$$q_{\text{b,tot,est},i} = \sum_{j=1}^{10} q_{\text{b,est},i,j} \tag{10}$$

Note that the exact same procedure was followed using the AH thresholds $th_{\text{ah},j}$ derived from Wyss et al. (2016a) (Eq. 1;
Table 3) to compare the performance between the AH method and the new AF method.
**3    Results**
**3.1    Flume experiments**
The flume experiments performed in the modified Avançon de Nant setup with the partition wall help to illustrate the two
calibration methods. Fig. 7a and 7b show the amplitude and frequency characteristics of all packets detected by the SPG
system during these experiments. Packets detected by the shielded sensor G1 all originate from impacts that occurred either
on the concrete bed or on plate G2 (Nicollier et al., 2021b). Packets detected by the unshielded sensor G2 are considered as
apparent if they are located in the area of the amplitude-frequency graph (Fig. 7a) where G1 and G2 packets overlap. Such
packets are presumed to have been triggered by impacts on the concrete bed too. The remaining packets, detected by G2 and
located in the non-overlapping area of the amplitude-frequency graph, are considered as real, rather than apparent. The
difference in $f_{\text{centroid}}$ between real and apparent packets (Fig. 7a) reflects the faster attenuation of higher frequencies during
wave propagation, as mentioned earlier. Size class boundaries derived by the AH method of Wyss et al. (2016a) encompass
all of the packets, both apparent and real (Fig. 7a). This is because the boundaries are defined solely by AH thresholds
($th_{\text{ah},j}$). By contrast, in the AF method proposed here, the two-dimensional class boundaries given by $th_{\text{af,low},j}$ and $th_{\text{af,up},j}$
cover only a fraction of all detected packets (Fig. 7b). Applying the step-like AF thresholds leads to a strong reduction of the
number of packets $PACK_j$ within each size class $j$ for plate G1 (shielded), particularly for the smaller classes. Meanwhile, the
AF thresholds had little effect on the number of detected packets for G2 (unshielded), except for a strong decrease for
classes $j = 1$ and 2, and a slight increase for classes $j = 6$ to 10 (Fig. 7c and 7d). Considering apparent packets as noise and
real packets as signal, applying the new AF method results in an increased signal to noise ratio, as shown by the larger
vertical separation between the blue (signal) and red (noise) lines in Fig. 7d compared to Fig. 7c.



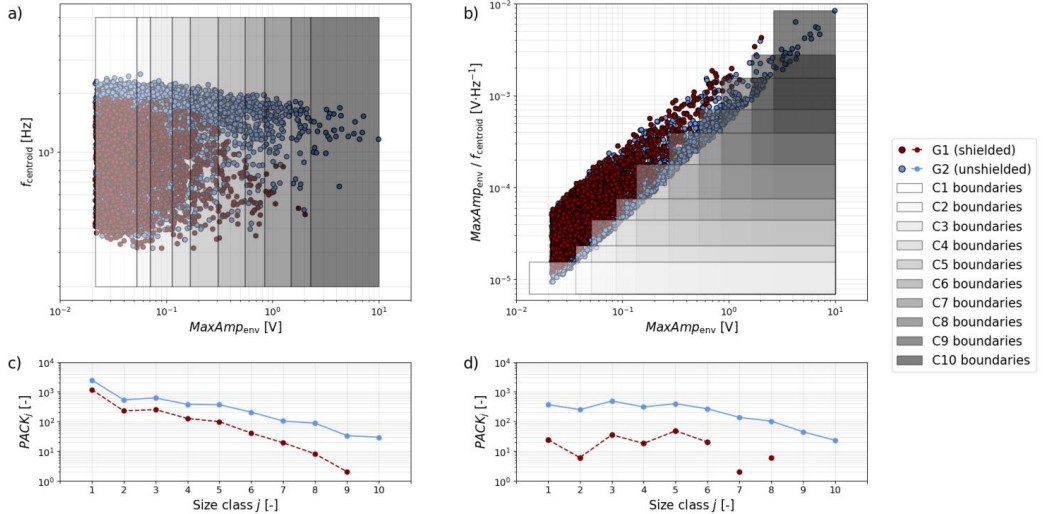

**Figure 7: Characteristics of the packets recorded during single-grain-size experiments conducted with the Avançon de Nant flume setup using the partition wall, with the maximum amplitude of the envelope $MaxAmp_{env}$ and the centroid frequency $f_{centroid}$. The red and blue dots correspond to packets recorded by the shielded plate G1 and the unshielded plate G2, respectively. The grey rectangles are the class boundaries delimited by the thresholds obtained for the AH method (a) and the AF method (b). In (a), $f_{centroid}$ is shown for information purposes only and is not incorporated in the thresholds. (c) and (d) represent the number of packets $PACK_j$ located within the class boundaries shown in (a) and (b), respectively. Missing markers signify that no packet was detected within the corresponding AH or AF thresholds.**

### 3.2 Field calibration coefficients

As discussed in the previous section, the number of packets $PACK_{i,j}$ detected for a given class $j$ varies together with the thresholds $th_{ah,j}$, $th_{af,low,j}$ and $th_{af,up,j}$. Because the measured fractional bedload mass $M_{meas,i,j}$ remains constant, the calibration coefficients $k_{b,i,j}$ will depend on the number of packets detected, and thus on the thresholds that are used to classify them. We can make the following observations regarding the calibration coefficients $k_{b,i,j}$ obtained using the AF method (Fig. 8b) compared to the AH method (Fig. 8a). First, the $k_{b,i,j}$ coefficients of the smaller size classes are substantially lower, meaning that fewer packets per unit mass are detected. Second, for the larger size classes, slightly more packets are detected per unit mass. Third, considering all sites and all size classes $j$, the overall scatter of the $k_{b,i,j}$ coefficients is smaller. This is reflected in the decrease of the mean coefficient of variation (CV) across all classes $j$ and all sites from CV = 1.17 (in the AH method) to CV = 0.93 (in the AF method). Fourth, the scatter of the site-specific $k_{b,i,j}$ coefficients is usually smaller. This is supported by the change of the mean CV across all classes from 0.89 to 0.54 for the Albula, from 0.83 to 0.75 for the Avançon de Nant and from 1.31 to 1.00 for the Erlenbach, between the AH and AF methods. The mean CV for the Navisence site however remains unchanged at 0.85. The general coefficients $k_{b,j,gen}$ obtained from the site-specific median coefficients $k_{b,j,med}$ using Eq. (7) are listed in Table 4.





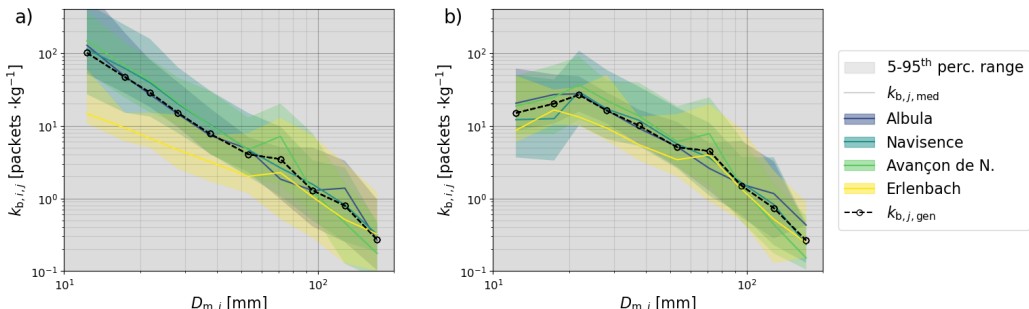

**Figure 8: The $k_{b,i,j}$ calibration coefficients obtained with the AH method (a) and the AF method (b) for each field site. The colored areas indicate the range between the 5th and the 95th percentile $k_{b,i,j}$ values, the full lines indicate the site-specific median coefficients $k_{b,j,med}$ and the black dashed lines indicate the final general calibration coefficients $k_{b,j,gen}$ as a function of the mean particle diameter $D_{m,j}$ of each grain-size class $j$.**

**Table 4: General calibration coefficients $k_{b,j,gen}$ obtained for each grain-size class $j$ with the AH method and the AF method using Eq. (7).**

|  | Method | Units | $j = 1$ | $j = 2$ | $j = 3$ | $j = 4$ | $j = 5$ | $j = 6$ | $j = 7$ | $j = 8$ | $j = 9$ | $j = 10$ |
|---|---|---|---|---|---|---|---|---|---|---|---|---|
| $k_{b,j,gen}$ | AH | kg$^{-1}$ | 100.67 | 46.43 | 28.68 | 15.03 | 7.76 | 4.04 | 3.47 | 1.29 | 0.79 | 0.27 |
|  | AF | kg$^{-1}$ | 14.97 | 20.15 | 26.65 | 16.15 | 10.06 | 5.05 | 4.49 | 1.50 | 0.74 | 0.27 |

### 3.3 Bedload flux estimates

We can now insert the general calibration coefficients $k_{b,j,gen}$ in Eq. (8) to compute fractional bedload mass estimates $M_{est,i,j}$ and subsequently the unit fractional flux estimates $q_{b,est,i,j}$ (Eq. 9) for every sample collected at the four field sites. Fig. 9 illustrates the accuracy of the bedload flux estimates obtained with the AF method for each sample across the grain-size classes and the field sites. The results obtained with the AH method can be found in Supplementary Information S3, and Table 5 provides further information on the performance of the two methods. The dashed colored power-law regression lines shown in Fig. 9, described by the corresponding linear coefficient $a$ and exponent $b$ (Table 5), indicate possible trends in over/under-estimation at each field site. The coefficient of determination $R^2$ describes the accuracy of the estimates relative to the 1:1 line. The root-mean-square error ($RMSE$) quantifies the expected error of the estimates and is expressed in [kg m$^{-1}$ s$^{-1}$]. When applied to the field calibration data, the AF method generally yields more accurate flux estimates than the AH method does. This is most notably reflected by the $R^2$ values and the percentages $p_{factor\_2}$ and $p_{factor\_5}$ of all detected samples whose estimated bedload fluxes differ by less than a factor of 2 and 5, respectively, from the measured values (Table 5). The five smallest grain-size classes were most strongly affected by these improvements, whereas the estimates for the largest fractions ($j = 7$ to $10$) were only slightly improved.

Aside from these comparative observations, it is also worth mentioning the following more general findings that are valid for both methods: (i) for most size fractions, the relative scatter of the estimates (on the log-log plots) decreases with increasing transport rates; (ii) at low transport rates, mass fluxes are generally overestimated, while at high transport rates they are generally underestimated; (iii) mass fluxes for the Erlenbach closely follow the 1:1 line but tend to be slightly underestimated; (iv) the number of measured ($N_{samples,meas}$) and estimated ($N_{samples,est}$) samples both decrease with



increasing particle size. Samples for which either the measured or the estimated flux equals 0 are indicated as dots along the
axes in Fig. 9. If the measured flux is zero but the estimated flux is positive, the sample can be regarded as false positive
(Fawcett, 2006). The difference between $N_{samples,meas}$ and $N_{samples,est}$ in Table 5 indicates that the occurrence of such false
positive samples increases with increasing particle size. Further performance metrics derived from the confusion matrix can
be found in the Supporting Information (Table S2).

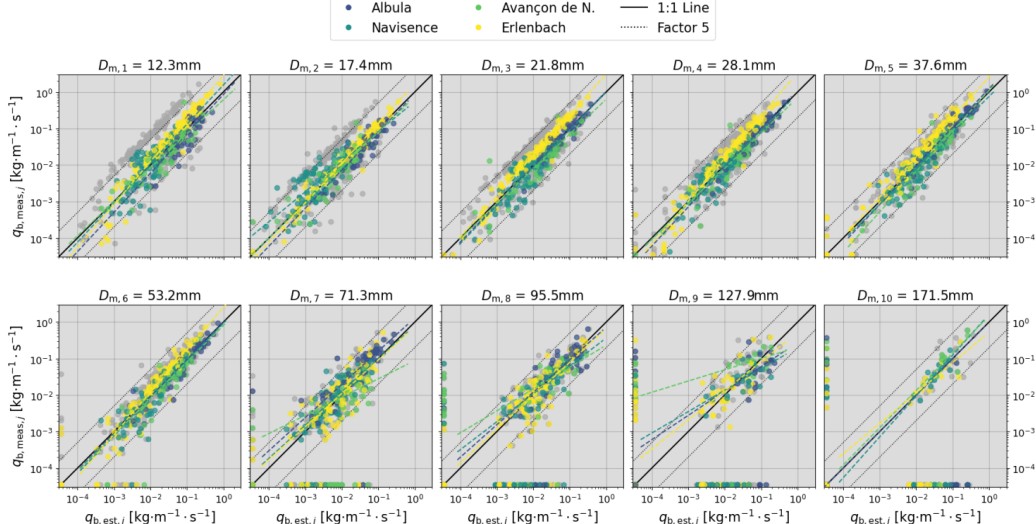


**Figure 9: Unit fractional transport rate estimates obtained with the AF method for each size class $j$ and each station. The light grey**
**dots in the background indicate the estimates obtained with the AH method and are represented in more detail in the Supporting**
**Information (Fig. S1). Each frame is annotated with the mean particle size $D_{m,j}$ of the represented class. The solid black lines**
**correspond to the reference 1:1 line while the dotted lines delimit factors of 5 above and below (from 0.2 to 5). The dashed colored**
**lines are power-law regression lines; the mean coefficients over all four sites are listed in Table 5. The dots along the axes indicate**
**samples for which either the measured or the estimated unit fractional flux equals 0. These samples are not considered for the**
**computation of the trend lines.**

**Table 5: Performance of the AH method and the AF method regarding fractional flux estimates for each class $j$ with following**
**parameters: the linear coefficient $a$, the exponent $b$ and the correlation coefficient $r$ of the power-law regression lines visible in Fig.**
**9; the coefficient of determination $R^2$; the root-mean-square error $RMSE$; and the percentage of all detected samples for which the**
**estimated value differs from the measured value by less than a factor of 2 and 5 $p_{factor\_2}$ and $p_{factor\_5}$, respectively. These values**
**were first computed for each site separately and then averaged over all four sites. The number of measured $N_{samples,meas}$ and the**
**number of estimated samples $N_{samples,est}$ showing a positive unit fractional rate were summed over all four sites.**

| | | Units | $j = 1$ | $j = 2$ | $j = 3$ | $j = 4$ | $j = 5$ | $j = 6$ | $j = 7$ | $j = 8$ | $j = 9$ | $j = 10$ |
|---|---|---|---|---|---|---|---|---|---|---|---|---|
| $N_{samples,meas}$ | | - | 308 | 308 | 306 | 306 | 302 | 287 | 240 | 213 | 112 | 53 |
| AH method | $N_{samples,est}$ | - | 308 | 305 | 307 | 301 | 299 | 289 | 267 | 237 | 149 | 117 |
| | $r$ | - | 0.77 | 0.83 | 0.87 | 0.88 | 0.91 | 0.89 | 0.73 | 0.75 | 0.53 | 0.46 |
| | $a$ | - | 3.6 | 2.02 | 1.95 | 2 | 1.39 | 1.54 | 0.85 | 0.53 | 0.42 | 0.58 |
| | $b$ | - | 0.94 | 0.95 | 1 | 1.05 | 1.01 | 1.05 | 0.83 | 0.83 | 0.64 | 0.6 |
| | $R^2$ | - | 0.4 | 0.51 | 0.64 | 0.70 | 0.78 | 0.81 | 0.36 | 0.57 | -0.16 | 0.11 |
| | $RMSE$ | kg·m$^{-1}$·s$^{-1}$ | 0.094 | 0.031 | 0.044 | 0.036 | 0.052 | 0.048 | 0.038 | 0.037 | 0.04 | 0.06 |
| | $p_{factor\_2}$ | % | 50 | 54 | 54 | 58 | 64 | 72 | 50 | 58 | 37 | 57 |
| | $p_{factor\_5}$ | % | 72 | 84 | 92 | 93 | 96 | 95 | 86 | 81 | 68 | 73 |





| | | | | | | | | | | | |
|---|---|---|---|---|---|---|---|---|---|---|---|
| | $N_{\text{samples,est}}$ | - | 308 | 305 | 307 | 305 | 301 | 295 | 279 | 242 | 161 | 84 |
| | $r$ | - | 0.79 | 0.82 | 0.89 | 0.91 | 0.93 | 0.93 | 0.81 | 0.78 | 0.52 | 0.61 |
| AF Method | $a$ | - | 1.46 | 0.96 | 1.44 | 1.54 | 1.41 | 1.3 | 0.73 | 0.49 | 0.3 | 1.16 |
| | $b$ | - | 1.07 | 0.98 | 1.03 | 1.05 | 1.06 | 1.05 | 0.81 | 0.79 | 0.59 | 0.74 |
| | $R^2$ | - | 0.71 | 0.72 | 0.8 | 0.84 | 0.85 | 0.83 | 0.42 | 0.55 | -0.08 | 0.59 |
| | $RMSE$ | kg·m$^{-1}$·s$^{-1}$ | 0.068 | 0.021 | 0.035 | 0.027 | 0.045 | 0.040 | 0.035 | 0.039 | 0.042 | 0.061 |
| | $p_{\text{factor\_2}}$ | % | 69 | 74 | 69 | 78 | 75 | 81 | 53 | 58 | 43 | 47 |
| | $p_{\text{factor\_5}}$ | % | 96 | 93 | 98 | 98 | 97 | 97 | 91 | 83 | 68 | 56 |


As indicated by Eq. (10), the unit total flux estimates are computed as the sum of the unit fractional flux estimates over
all 10 classes. Fig. 10 shows the ratio $r_{q_{b,\text{tot}}}$ between the estimated total flux $q_{b,\text{tot,est}}$ and the measured total flux $q_{b,\text{tot,meas}}$
for all 308 calibration samples, as a function of the sampled total mass $M_{\text{tot,meas}}$. Here, the estimates for the Albula, the
Navisence and the Avançon de Nant sites are slightly more accurate with the AF method than with the AH method, whereas
the estimates for the Erlenbach improve substantially, with the median $r_{q_{b,\text{tot}}}$ value increasing from 0.31 to 0.64. Note that
the observations (i) to (iii) made earlier regarding the fractional flux estimates are also valid here. Fig. 10 also provides an
interesting overview of the sampled masses at all four stations, reflecting the capacities of the different devices (automated
and manual basket samplers and crane-mounted net sampler) used to collect the calibration samples.

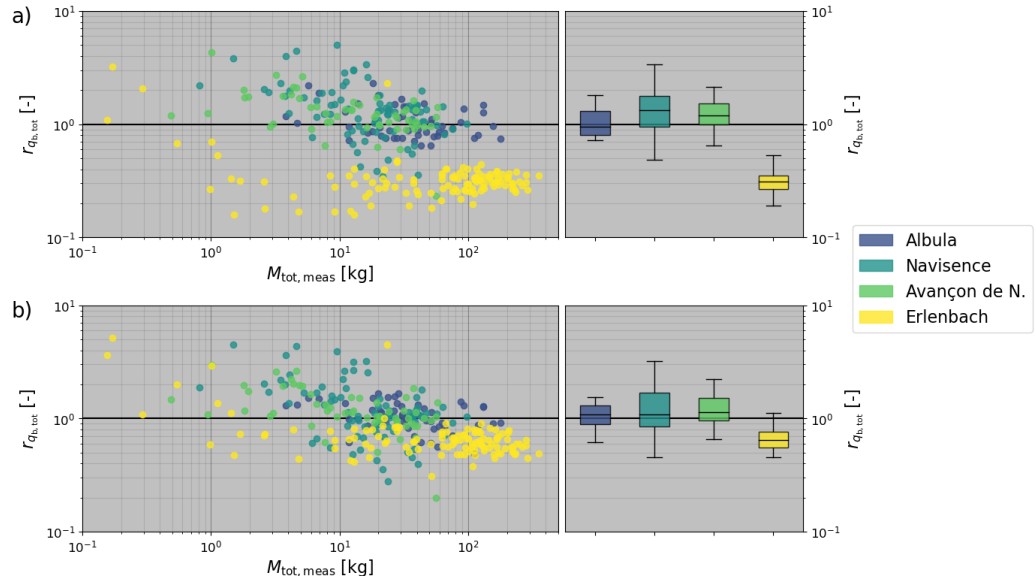


**Figure 10: Ratio $r_{q_{b,\text{tot}}}$ between the estimated and the measured unit total mass flux as a function of the total sampled mass**
**$M_{\text{tot,meas}}$, for each collected sample $i$ and each station, for the AH method (a) and the AF method (b). The boxplots on the right**
**indicate the range of $r_{q_{b,\text{tot}}}$ values obtained for each station.**
**3.4   Grain-size estimates**
We can combine the SPG bedload flux estimates for all grain-size fractions and thus derive grain-size distributions, which
can then be compared to the measured size distributions of each calibration sample. Fig. 11 compares the performance of the
AH and the AF methods in estimating the characteristic grain sizes $D_{30}$, $D_{50}$, $D_{67}$ and $D_{84}$ (where $D_x$ is the grain diameter for
which $x$ percent of the sampled bedload mass is finer). The accuracy of the estimates is indicated by the ratio $r_{D_x}$ between the



estimated and the measured characteristic grain size $D_x$. Compared to the AH method, the AF method mainly improves the
estimates of the four characteristic grain sizes for the Navisence and the Erlenbach sites, but has little effect at the other two
sites. The largest improvement is achieved for the Erlenbach site, with the median $r_{D_{30}}$ changing from 1.37 to 1.02, the
median $r_{D_{50}}$ changing from 1.48 to 1.01, the median $r_{D_{67}}$ changing from 1.46 to 1.05 and the median $r_{D_{84}}$ changing from
1.39 to 1.10. The overall accuracy of the estimates decreases with increasing characteristic size $D_x$ for both methods, and for
every characteristic size $D_x$, the $D_x$ tends to be overestimated for finer grain mixtures and underestimated for coarser grain
mixtures.

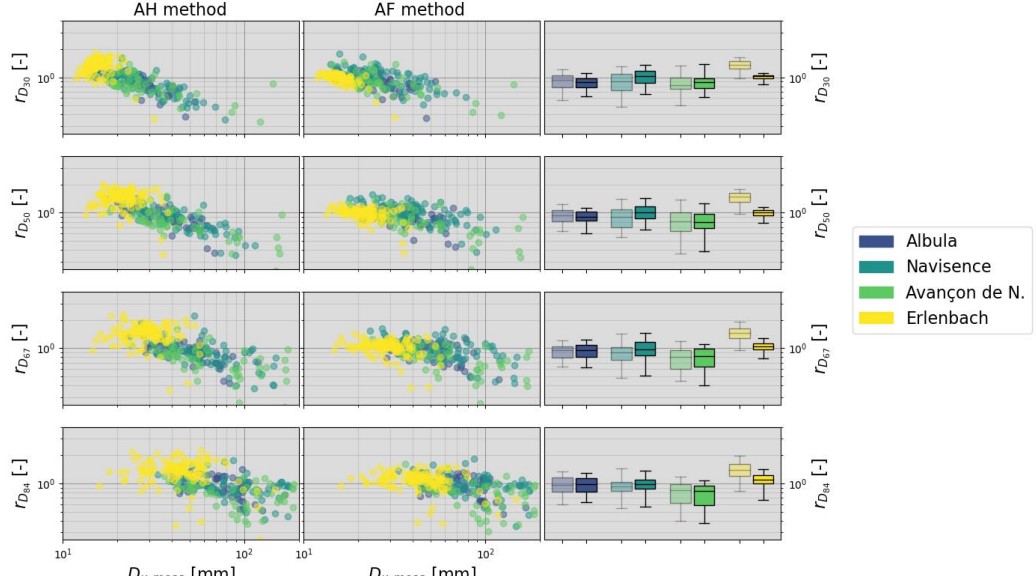


**Figure 11: Ratio $r_{D_x}$ between the estimated and the measured characteristic grain sizes $D_{30}$, $D_{50}$, $D_{67}$ and $D_{84}$ as a function of the**
**measured grain diameter $D_{x,\text{meas}}$ for each collected sample $i$ and each station using the AH method (column 1) and the AF method**
**(column 2). $D_x$ is the grain diameter for which $x$ percent of the sampled bedload is finer. The boxplots in column 3 indicate the**
**range of $r_{D_x}$ values obtained for each station. The boxes in faded colors show the results obtained with the AH method and the**
**boxes in brighter colors show the results obtained with the AF method.**
**4    Discussion**
**4.1    The hybrid calibration procedure**
Recent studies have pointed out the difficulty of transferring flume-based calibrations of the impact plate system to field
applications (e.g. Mao et al., 2016; Wyss et al., 2016c; Kuhnle et al., 2017). In the hybrid calibration approach presented
here, we took advantage of controlled flume experiments, but only to obtain amplitude and amplitude-frequency thresholds
for each particle-size class, which were subsequently applied to field calibration datasets to derive the general calibration
coefficients $k_{b,j,\text{gen}}$.
Among the three sites reconstructed at the flume facility, only the experiments conducted in 2018 with the Albula setup
were used for calibration purposes in the present study. Although the differences are small, the class thresholds derived from
these experiments yielded slightly more accurate bedload flux estimates than the thresholds derived from the other site





reconstructions. A possible explanation for this is the lower bed roughness used for the Albula site reconstruction as
compared to the other two setups, which facilitated the transport of larger particles. The Albula setup was also less affected
by lateral sorting of small particles (mainly classes $j = 1$ to 4) toward the flume walls, which resulted in a weaker signal
response. Additionally, the flow velocities used in this setup ($V_\mathrm{f} = 1.6$ and 2.4 m s$^{-1}$) lie between the velocities measured
during the field calibration campaigns at the Navisence and Avançon de Nant sites.

The entire hybrid calibration procedure was run iteratively until the optimal linear coefficient and exponent of the
criterion (Eq. 3) used to filter out apparent packets were found (Fig. 6). As objective function we used an equally weighted
combination of parameters describing the accuracy of bedload flux and grain-size estimates, i.e. $r$, $R^2$, $p_{\mathrm{factor}\_2}$, $p_{\mathrm{factor}\_5}$, and
$RMSE$ as shown in Table 5, $r_{D_x}$ as shown in Fig. 11, and the accuracy derived from the confusion matrix (Fawcett, 2006) as
shown in Table S2 in Supporting Information. We looked for two types of optimal calibrations. The first type is a general
calibration, for which we have presented the results in Sect. 3. This calibration combines all four stations in order to
investigate the feasibility of a general signal conversion procedure applicable to multiple sites equipped with SPG systems.
The second type is a site-specific calibration aiming to improve the accuracy of bedload transport rate estimates at a single
monitoring station, to be used for a more detailed analysis of bedload-related processes at a given site (details of these site-
specific calibrations are available in Supporting Information Sect. S4 and S5).

The biases introduced by apparent packets can be removed by site-specific calibration of the coefficients $k_{\mathrm{b},i,j}$, so the
AF and AH methods perform about equally well when calibrated separately to each individual site (see Supporting
Information Sect. S4 and S5). However, the abundance of apparent packets varies considerably from site to site, owing to
differences in the channel geometry, the bedload grain-size distribution, and the construction details of the individual SPG
installations. Because the AF method filters out a substantial fraction of these apparent packets, it yields substantially better
general calibrations than the AH method does (see Table 5).

We also tested the performance of an adapted version of the AH method introduced by Rickenmann et al. (2018). This
method was originally developed for the Erlenbach site and aimed to correct for the relationship between the signal response
and the transport rate. In the present study, we applied this method to each field site. The only notable improvement
introduced by the adapted AH method is the increased number of detected samples at the Erlenbach station, leading to more
accurate estimates of the various characteristic grain sizes $D_x$ at this site (Tables S8 and S9 in Supporting Information); the
results for the other sites were not substantially improved.
**4.2   Two-dimensional size class thresholds**
To understand the performance of the new AF method it is worth taking a closer look at the role of the size class thresholds.
As shown in Fig. 7, replacing the upper amplitude thresholds with amplitude-frequency values results in the following two
important changes. First, a dimension is added, which facilitates focusing on the narrow range of signal responses
characteristic for real packets, and filtering out many of the apparent packets. Second, the areas of the amplitude-frequency
domain covered by two adjacent classes can now overlap. Packets located in overlapping areas are assigned once to each
class and therefore counted twice. This explains why both the number of detected packets $PACK_j$ (Fig. 7c and 7d) and
subsequently the $k_{\mathrm{b},j}$ values (Fig. 8) are slightly higher when the AF method (instead of the AH method) is applied to the
larger size classes. Counting such packets twice is not unreasonable, given that the ranges of signal responses recorded
during single-grain-size flume experiments for two contiguous grain-size classes significantly overlap, even after apparent
packets are filtered out (Fig. 5).

Through the reduced area covered by the new amplitude-frequency thresholds in Fig. 7b, a certain percentage of all the
packets recorded during the field calibration experiments is neglected for general calibration: 55% at the Albula site, 63% at
Navisence, 58% at Avançon de Nant and only 9% at Erlenbach. This suggests that the plates embedded at Erlenbach pick up



less noise from their surroundings. A similar trend was observed by Nicollier et al. (2021b) when comparing the maximum
amplitude registered by two adjacent plates for a given impact at the same location. This difference in noise detection levels
is possibly accentuated by the number of impacted plates during bedload transport events. The SPG array embedded in the
artificial U-shaped channel of the Erlenbach has the particularity that only 2 out of its 12 plates are usually impacted by
bedload particles during floods (and only sediment crossing these two plates is caught by the automatic basket sampler),
while at the other stations all 10 to 30 embedded plates are submerged by the flow and thus can potentially be impacted.

## 4.3 Sampling uncertainties

Even though the AF method improved the overall accuracy of flux estimates for most classes (Table 5), some trends
addressed in Sect. 3 suggest that factors other than the noise level also control the accuracy of the estimates. The dataset
presented in this study includes 308 calibration measurements and is in our knowledge the largest dataset gathered for an
impact plate system. Still, it appears that the number of collected samples is not sufficient to accurately assess the
performance of the two methods for the three largest particle-size classes (Fig. 9; Table 5). This is mainly due to the fact that
in typical sediment mixtures, large particles are rarer than fine particles (Rickenmann et al., 2014; Mao et al., 2016). Earlier
investigations have shown that a larger number of detected bedload particles reduces the scatter of total mass estimates by
averaging over stochastic factors such as the impact location on a given impact plate, the particle transport mode (sliding,
rolling, saltating, etc.), and the impact velocity (Rickenmann and McArdell, 2008; Turowski et al., 2013). A further
uncertainty arises because these larger particles are transported at higher bed shear stresses (Einstein, 1950; Wilcock and
Crowe, 2003), which also mobilize more total material and thus pose a serious challenge regarding the sampling efficiency
of the calibration bedload samplers. Bunte and Abt (2005) and Bunte et al. (2019) have demonstrated that reducing the
sampling duration with a bedload trap from 60 to 2 minutes decreases both the sampled unit total bedload flux $q_{b,tot}$ and the
sampled maximum particle size $D_{max}$ by about half. In the present study, total bedload fluxes up to 4 kg m$^{-1}$ s$^{-1}$ were measured
with the net sampler, meaning that the measurement duration had to be minimized to avoid overloading the sampler. At the
Albula stream, for instance, only four samples contained particles of the largest class, and all four were sampled over a
duration ranging from 1 to 2 minutes. As a comparison, the longest sampling duration was reached at the Navisence site and
lasted 25 minutes. All this suggests that an optimal calibration of the SPG system requires balancing the sampling duration
and the number of collected particles. Flume experiments could potentially be used to assess the sampling efficiency of the
various calibration sampling methods, along with the detection efficiency of the SPG system.

## 4.4 Transport rate

Two further trends are evident in the unit fractional flux estimates obtained for the seven smallest classes, for which most
samples were detected ($N_{samples,est}$ / $N_{samples,meas}$ > 96%; Table 5). First, the relative scatter (on the log-log plots) of the
fractional flux estimates around the power-law regression lines in Fig. 9 is smaller at higher transport rates. Second, both
total and fractional fluxes are generally overestimated at low transport rates and underestimated at high transport rates (Fig. 9
and 10), which also correspond to the largest calibration samples. These findings agree with results from previous calibration
campaigns with the SPG system (Rickenmann and Fritschi, 2017; Rickenmann et al., 2018) but a comprehensive explanation
for these trends is still missing. The following hypotheses can be put forward to explain the relationship between the mass
flux estimates and the transport rate $q_b$: (i) The SPG system may suffer from saturation when the transport rate is too high, as
has been document in the Japanese pipe microphone system (Mizuyama et al., 2011; Choi, 2020). In our SPG data, we have
observed long packets containing multiple large peaks corresponding to several impacts occurring so quickly after one
another that they were not detected as separate packets. One can expect that the probability of occurrence of such packets





increases together with the transport rate, with the transport of large particles (which typically generate packets of longer durations), and with the occurrence of sliding and rolling particles (Chen et al., 2021). The long packets take the place of multiple shorter packets that would otherwise be individually counted; thus, they lead to underestimated mass fluxes for a given $k_{b,j}$ value. (ii) Field observations of bedload sheets being transported over plates at high transport rates were made at the Vallon de Nant site. In the presence of bedload sheets, one can expect that the detection rate of transported particles is hampered by multiple particle layers (Rickenmann et al. 1997; Turowski and Rickenmann, 2009), kinetic sieving (e.g. Frey and Church, 2011) or percolation processes (e.g. Recking et al., 2009).

We are not able to give a clear explanation for the overestimates of the characteristic grain size $D_x$ for finer grain mixtures and underestimates for coarser grain mixtures (as shown in Fig. 11). A similar trend was also observed by Rickenmann et al. (2018) for calibration measurements originating from the Erlenbach. We speculate that the decrease of the detection rate along with increasing transport intensity, as mentioned above, may partly explain this phenomenon.

## 4.5 Effect of the flow velocity

A recurrent feature in the results presented above is an offset between the estimates obtained for the Erlenbach and those obtained for the three other stations. A similar offset was observed earlier for linear calibration relations for total bedload mass between the Erlenbach and other field sites with more natural approach flow conditions (Rickenmann et al., 2014). Although applying the new amplitude-frequency method has reduced the offset in the present study significantly, it remains visible for both fractional and total bedload flux estimates (Fig. 9, 10, and 12). At the Erlenbach site, the last 35 meters upstream of the SPG system consist of an artificial bed with a steep channel slope of 16%, consisting of large flat embedded boulders (Roth et al., 2016). This explains the supercritical flow regime with a Froude number around 5.1 (Wyss et al., 2016c) and a flow velocity $V_f$ around 5 m s$^{-1}$ at the check dam with the geophone sensors (Table S1). Bedload particle velocity $V_p$ was introduced by Wyss et al. (2016c) as a possible governing parameter affecting the number of particles detected by the SPG system. For the present study, we used $V_f$ as a proxy for $V_p$, even though bedload particles generally travel more slowly than the fluid that surrounds them (Ancey et al., 2008; Chatanantavet et al., 2013; Auel et al., 2017). Past flume experiments (Wyss et al., 2016a; Kuhnle et al., 2017) have shown that the calibration coefficient $k_{b,j}$ can vary with the flow velocity $V_f$, such that a three-fold increase in $V_f$ can lead to a two-fold decrease of $k_{b,j}$. Furthermore, bed morphology, bed roughness and flow velocity play important roles in determining particle transport mode, i.e., sliding, rolling, or saltating (e.g. Bagnold, 1973; Lajeunesse et al., 2010). Although high flow velocities generally favor the saltating mode (Ancey et al., 2002), the shallow flow depths measured at the Erlenbach (in average 10 cm; Wyss et al. 2016b) may limit the hop height of larger particles (Amir et al., 2017). Considering all these aspects, we hypothesize that the generally underestimated transport rates observed for the Erlenbach site mainly arise from the exceptionally high flow velocity and the related transport mode (Fig. 12). Continuous flow velocity measurements are lacking at the Albula and Navisence sites, hampering a more detailed analysis of their relationships between flow velocities and detection rates.



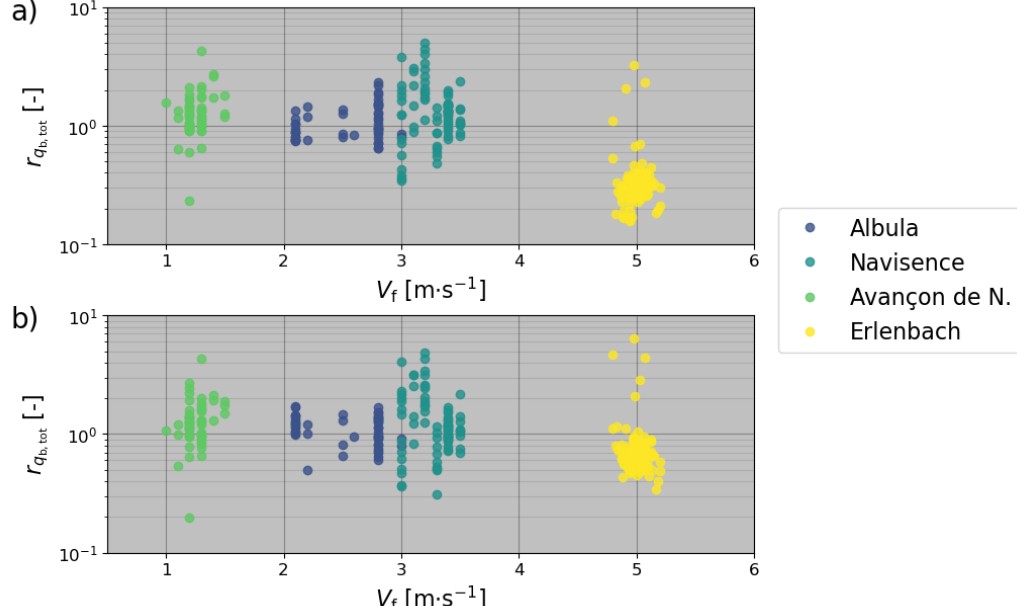

553

**Figure 12: Ratio $r_{q_{b,tot}}$ between the estimated and the measured unit total mass flux as a function of the mean flow velocity $V_f$, for each collected sample and each station, for the AH method (a) and the AF method (b). The indicated flow velocity corresponds to in situ measurements made during (or close in time to) the corresponding calibration measurement. For better readability, a random scatter ranging from -0.2 m s$^{-1}$ to 0.2 m s$^{-1}$ was added to the stable flow velocity of 5 m s$^{-1}$ measured at the Erlenbach site.**

### 4.6 K-fold cross-validation

In a last stage, we tested the robustness of the AH and AF methods by splitting the dataset into calibration and validation data. Given that the number of calibration measurements is relatively small and varies between stations, we applied a 4-fold cross-validation technique (e.g. Khosravi et al., 2020). The field calibration measurements were distributed over four folds, each containing an equal number of calibration measurements from each site (Supporting Information Fig. S4). One after another, the folds were used as validation datasets while the remaining three folds were used for calibration. General calibration coefficients $k_{b,j,gen}$ were obtained from the calibration dataset and subsequently applied to the validation data to derive flux estimates. Even though each fold contains a total of only 48 samples (12 per site), the results obtained with the 4-fold cross-validation procedure support our conclusion that including frequency information in the packet classification procedure improves the mean accuracy of the estimates over all sites, in particular for the smaller five to six size classes $j$ (Supporting Information Table S10). Nicollier et al. (2021b) found that most apparent packets are detected as belonging to smaller size classes than the particles that caused them, due to the attenuation of the vibrations as they propagate (see Fig. 7). It is therefore reasonable that the AF method mainly improves the flux estimates for these smaller classes.

### 5 Conclusion

The Swiss plate geophone (SPG) is a bedload surrogate monitoring system that has been installed in several gravel-bed streams and was calibrated using direct sampling techniques. While most site-specific calibration relationships for total mass flux are robust across several orders of magnitude, the mean calibration coefficients can still vary by about a factor of six between different sites. In this study, we derived a general procedure to convert SPG signals into fractional bedload fluxes



using an extensive dataset comprising controlled flume experiments as well as 308 field calibration measurements from four
field sites. The proposed hybrid approach is based on previous findings (Antoniazza et al., 2020; Nicollier et al., 2021b) that
the SPG system is biased by elastic waves that propagate through the apparatus and generate noise in the form of spurious
"apparent" packets. We introduced the amplitude-frequency (AF) method as an alternative to the amplitude-histogram (AH)
method developed by Wyss et al. (2016a). Packets recorded during single-grain-size flume experiments were first filtered to
exclude apparent packets, and then used to derive grain-size class thresholds for packet classification. We found that filtering
out apparent packets results in more consistent relationships between particle diameter and amplitude-frequency
characteristics of the SPG signal. Furthermore, we showed that including frequency information in size class thresholds
helps in excluding apparent packets and thus improves the signal-to-noise ratio. In a second stage, we applied these flume-
based thresholds to field calibration measurements and derived general calibration coefficients applicable at all four sites for
ten different grain-size fractions. The AH method, by contrast, requires site-specific calibration because it cannot account for
the site-to-site differences in the abundance of apparent packets. Averaged over the ten grain-size fractions, the bedload
masses of 69% and 96% of the samples were estimated within an offset of a factor of two and five, respectively, relative to
the measured sampled masses. The remaining discrepancies between the site-specific results are mainly attributed to large
differences in flow (and probably particle) velocity. Finally, the sampled mass, the transport rate and the sampling efficiency
were identified as further factors possibly influencing the accuracy of mass flux and grain-size estimates.
The presented results are highly encouraging regarding future applications of surrogate monitoring methods to
investigate bedload transport processes. The findings also underline the valuable contribution of flume experiments to our
understanding of the relationship between bedload transport and the recorded SPG signal. But above all, this study highlights
the requirements for obtaining calibrations that are transferable across sites: accurate and numerous direct sampling
measurements with long sampling durations and large sampled masses, sensors insulated from surrounding noise sources,
and highly resolved temporal information about the stream flow, to identify and account for variations in the transport
conditions.
**Notation**
$a_c$  Linear coefficient of the criterion
$A_{FFT}$  Fourier amplitude
$A_{m,j}$  Mean amplitude registered for particle-size class $j$
$b_c$  Linear coefficient of the criterion
$\Delta t_i$  Sampling duration
$D_{m,j}$  Mean particle diameter for particle-size class $j$
$D_{sieve,j}$  Lower sieve size retaining particle class $j$
$D_x$  Characteristic grain size
$f_{centroid}$  Centroid frequency
$i$  Sample index
$j$  Particle-size class index
$k_{b,i,j}$  Sample- and class-specific calibration coefficient
$k_{b,j,med,station}$  Median calibration coefficient for particle-size class $j$ and a given station
$k_{b,j,gen}$  General calibration coefficient for particle-size class $j$
$M_{est,i,j}$  Estimated fractional mass per sample and per class
$M_{meas,i,j}$  Sampled fractional mass per sample and per class
$MaxAmp_{env}$  Maximum registered amplitude within a packet



| 617 | $N_{\text{samples,est}}$ | Number of detected samples |
|-----|-----|-----|
| 618 | $N_{stations}$ | Number of stations |
| 619 | $PACK_{i,j}$ | Number of recorded packets per sample and per class |
| 620 | $p_{\text{factor\_x}}$ | Percentage of all detected samples for which the estimated and the measured values differ from each |
| 621 | | other by less than a factor of $x$ |
| 622 | $q_{\text{b,est},i,j}$ | Estimated unit fractional transport rate per sample and per class |
| 623 | $q_{\text{b,meas},i,j}$ | Measured unit fractional transport rate per sample and per class |
| 624 | $q_{\text{b,tot,est},i}$ | Estimated unit total bedload flux per sample |
| 625 | $q_{\text{b,tot,meas},i}$ | Measured unit total bedload flux per sample |
| 626 | $R^2$ | Coefficient of determination |
| 627 | $r$ | Correlation coefficient |
| 628 | $r_x$ | Ratio between estimated and measured values $x$ |
| 629 | $th_{\text{ah},j}$ | Amplitude-histogram thresholds |
| 630 | $th_{\text{af,low},j}$ | Lower amplitude-frequency thresholds |
| 631 | $th_{\text{af,up},j}$ | Upper amplitude-frequency thresholds |
| 632 | $V_{\text{f}}$ | Mean flow velocity |
| 633 | $w_{\text{p}}$ | Standard width of an impact plate |

**Data availability**
The dataset presented in this paper is available online on the EnviDat repository
https://www.envidat.ch/#/metadata/sediment-transport-observations-in-swiss-mountain-streams.
**Author contribution**
Tobias Nicollier designed and carried out the field and flume experiments, developed the presented workflow and prepared
the manuscript with contributions from all co-authors. Gilles Antoniazza designed and carried out the field experiments at
the Vallon de Nant site. Lorenz Ammann helped developing the methodology and contributed to the formal analysis. Dieter
Rickenmann contributed to the conceptualization and the supervision of the presented work, contributed to the design of the
methodology, and provided support during the field and flume experiments. James W. Kirchner contributed to the
development of the methodology and significantly contributed to the preparation of the initial draft.
**Acknowledgements**
This study was supported by Swiss National Science Foundation (SNSF) grant 200021L_172606, and by Deutsche
Forschungsgemeinschaft (DFG) grant RU 1546/7-1. The authors are grateful to Arnd Hartlieb, to the students of the TU
Munich, and to the technical staff of the Oskar von Miller Institute for helping to set up and perform the flume experiments.
They also warmly thank Norina Andres, Mehdi Mattou, Nicolas Steeb, Florian Schläfli, Konrad Eppel and Jonas von
Wartburg for their efforts and motivation during the field calibration campaigns. Special thanks go to Andreas Schmucki,
who never gave up repairing the net sampler. Alexandre Badoux is further thanked for his valuable suggestions regarding an
earlier version of the manuscript.



**Competing interests**
The authors declare that they have no conflict of interest.

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
