# Peer review of "Toward a general calibration of the Swiss plate geophone system for fractional bedload transport"

_Earth Surface Dynamics, 2022_

## Referee Comment (RC1)

Review of "Toward a general calibration of the Swiss plate geophone system for fractional bedload transport" by T Nicollier et al

**CONTRIBUTIONS AND AUDIENCE**

This paper is a new contribution of the WSL team on the Swiss Plate geophone measurement technic. The author propose a method of analysis accounting for elastic waves resulting from impacts occurring outside the plate boundaries, with the final objective being to propose a general site-independent calibration procedure. They use both flume and field observations. This contribution will undoubtedly be of interest to the entire community using this technique.

The paper is well written and the science is of good quality. However I found the paper a bit long and sometimes difficult to understand. I made a few proposition to improve the text. I propose minor revision.

**COMMENTS**

Fig1: in the preprocessed signal do you record a value for each threshold or only the maximum?

Line 116: what was the mesh size for direct sampling? And related question: what is the size minimum detected by the SPG?

Line 118: a bedload sample is a mass collected for a given duration. It means that the corresponding "packet" is not the response of a single grain impact but probably a complex signal resulting from many impacts (or even a bedload pulse response)?

Line 119: what are the signification of the different letters in $k_{b,i,j}$?

Table 1: how was measured the flow velocity (surface?)

Line 146: you mean "uniform mixture"?

Line 173-174: what happens when several grains hit the plate simultaneously? (the question concerns SPG in the field)

Table3: it is very difficult to understand this table and its title

Line 180: Is this equation is site specific? First the material: can we consider that all sensors have exactly the same response? Secondly the bedload data may be specific to Erlenbach (mode of transport, grains velocity, density…)?

Line 183: I don't understand. You use the measured packets with Eq.1 for computing each size class present in a bedload mixture?

Line 200: why "in the lower size class"? I would expect that only large particles produce extraneous impacts?

Line 234-235: "The transported bedload mass associated with an individual signal packet is strongly dependent on the size of the impacting particle" what is difficult with such a sentence is that we don't really understand if you describe the movement of a single particle or of a bedload mixture..

Line 237: hard to follow. If I understood well you will apply a threshold to both amplitude and frequency. In the next sentence "lower threshold" and "upper threshold" concerns amplitude or frequency?

Line 250: could you tell a bit more about these coefficients?

Line 255: where do these equations come from? your experiments?

Line 257: If I understood, by replacing Dmj (the sieve sizes) in Eq 4 and 5 the objective is to isolate the packets associated with a given size class? Not clear (same for figure 5)

Line 279: you must imagine that you present to somebody who knows nothing about your work. Since I am reading, I am still lost with your upper and lower threshold.

Line 282: YES!! I have my answer!!

Line 284: The link between Eq.6 and 7 is not clear (I suppose that med station refers to all samples i)

Line 287: providing a general methodology for reducing the measurement uncertainties in a given site is already a nice objective. But the passage to a general inter-site calibration term is not trivial. It supposes that beside the plate response, all sites share the same transport characteristics. For instance if grains saltate over long distances (and different station length form one river to another) can we be sure that the impact rate reflect the real transport?

Line 346: the scatter is lower? Not so clear in the figure

Figure 9: add the light grey dots in the legend

Line 407: Direct sampling depends on the mesh size and the SPG measurements concern sizes >12mm. We know that in many mountain streams the contribution of gravels and sand can be very large. How can you take this into account?

Figure 10: How do you explain a small tendency to overprediction for lower transport?

Line 405: it questions on the pertinence of a general calibration coefficient. Also, many sites are equipped with SPG. Could it be possible to test the calibration coefficient with other sites?

§4.1: the paper is already very long and not easy to read. Is this paragraph really necessary? Or maybe to be reduced..

Line 484: It partly answer to see my previous comment about limitations of a general calibration

§ 488: Huge question which also concerns the contribution of finer fraction, how the saltation length of large elements affect the SPG detection…

Line 519: "In our SPG data, we have observed long packets containing multiple large peaks corresponding to several impacts occurring so quickly after one another that they were not detected as separate packets". It's a shame that this comment appears at the end of the manuscript because it's the image we immediately have in mind, which doesn't match the definition of the package in Figure1. It could be worth to explain early how you considers this aspect in your analysis.

§4.5 same comment. From the beginning I suspect grains velocity to play a role in the SPG response. I regret that this parameter is totally occulted in the paper. Even if you do not consider it in the analysis, it could be introduced earlier.

---

## Author Comment (AC1)

**Response to the comments made by Referee #1**

Dear Referee #1,

We appreciate your valuable comments and the several points you have raised regarding key elements of the manuscript. We also very much appreciate that you went through the study in such a thorough manner. We believe that your comments, questions and suggestions significantly helped to improve it. We agreed with most of your suggestions, and have made the modifications accordingly. Below, your comments are reported in italics, and our responses in normal font (blue color). The indicated line numbers refer to the tracked-changes version of the revised manuscript.

**Comment 1:** *This paper is a new contribution of the WSL team on the Swiss Plate geophone measurement technic. The author propose a method of analysis accounting for elastic waves resulting from impacts occurring outside the plate boundaries, with the final objective being to propose a general site-independent calibration procedure. They use both flume and field observations. This contribution will undoubtedly be of interest to the entire community using this technique.*
*The paper is well written and the science is of good quality. However I found the paper a bit long and sometimes difficult to understand. I made a few proposition to improve the text. I propose minor revision.*

Response: We kindly thank you for your positive comments on our work. We agree that the paper was quite long and that the clarity could be improved. As described in the following answers to your comments, we focused in particular on Section 2.5.2 (Lines 252-303) in order to clarify the procedure leading to the lower and upper threshold values. Another key element is certainly Figure 7. In order to clarify the effect of the new thresholds when applied to SPG data, we have included a short quantitative analysis of Figure 7 to Section 3.1 (Lines 338-369).

**Comment 2:** *Fig1: in the preprocessed signal do you record a value for each threshold or only the maximum?*

Response: While this study focuses on packets, most earlier publications on the SPG system were using impulse counts as proxy for bedload transport. In the preprocessed signal, we store the total number of packets or impulses detected for each class $j$ within one minute (defined by the threshold amplitudes). Packets are classified and counted on the basis of their maximum amplitude (i.e. one unique amplitude value is recorded per packet), and impulses, which are discrete points in time, simply on the basis of their amplitude. However, for this study, we used the full raw data that was recorded for all calibration measurements. We have changed the formulation in the first paragraph of section 2.1 (Lines 100-106), to put less emphasis on the preprocessed signal.

**Comment 3:** *Line 116: what was the mesh size for direct sampling? And related question: what is the size minimum detected by the SPG?*

Response: The lower size detection threshold is assumed to be around 10 mm. The lower amplitude threshold of 0.0216 V corresponds to a particle size of 9.5 mm. We have added two sentences clarifying this on Lines 96-97 and 128. From flume experiments, however, we know that particles of that size generate only few packets per unit mass (Wyss et al., 2016a; Nicollier et al. 2021). This limitation is probably related to the important mass of a steel plate.

We have therefore chosen the mesh size of the nets used to collect samples accordingly to this detection threshold. The mesh size was 8 mm at the Albula and the Navisence, and 6 mm at the Avançon de Nant site, where the flow was much weaker. One reason for choosing such large mesh sizes was to reduce the flow resistance of the net sampler in order to improve the sampling efficiency. We have added the mesh size information in Section S1 of the Supporting Information, where the calibration measurements are described in more details.

- Wyss, C. R., Rickenmann, D., Fritschi, B., Turowski, J., Weitbrecht, V., and Boes, R.: Measuring bed load transport rates by grain-size fraction using the Swiss plate geophone signal at the Erlenbach, *J. Hydraul. Eng.*, *142*(5), https://doi.org/10.1061/(ASCE)HY.1943-7900.0001090,04016003, 2016a.

- Nicollier, T., Rickenmann, D., and Hartlieb, A.: Field and flume measurements with the impact plate: Effect of bedload grain-size distribution on signal response, *Earth Surf. Processes Landforms*, 17 pp., https://doi.org/10.1002/esp.5117, 2021.

**Comment 4:** *Line 118: a bedload sample is a mass collected for a given duration. It means that the corresponding "packet" is not the response of a single grain impact but probably a complex signal resulting from many impacts (or even a bedload pulse response)?*

Response: In most cases, because of the high sampling frequency of the geophone signal (10'000 Hz), one packet contains the signal response corresponding to one single particle impact only. As discussed in Sect. 4.4, we hypothesize that at higher transport rates, the frequency of packets containing the signal response to more than one impact is expect to increase, and thus there may be some overlapping recording of packets. In future work, it could be worth investigating if splitting up such longer packets in sub-packets, each containing the signal responses to one single impact, results in different calibration coefficients.

**Comment 5:** *Line 119: what are the signification of the different letters in $k_{b,i,j}$?*

Response: The letter b is the standard subscript of the calibration coefficient (e.g. Wyss et al., 2016a), the letter *i* stands for the sample's index, and the letter *j* stands for the size class. We have noticed that it is certainly too early to introduce the calibration coefficient $k_{b,i,j}$ at this stage, since it appears only later in the text,

when describing Eq. 6. In order to avoid any confusion, we have removed the variable and kept only "calibration coefficient" (see Line 126).

- Wyss, C. R., Rickenmann, D., Fritschi, B., Turowski, J., Weitbrecht, V., and Boes, R.: Measuring bed load transport rates by grain-size fraction using the Swiss plate geophone signal at the Erlenbach, *J. Hydraul. Eng.*, *142*(5), https://doi.org/10.1061/(ASCE)HY.1943-7900.0001090,04016003, 2016a.

**Comment 6:** *Table 1: how was measured the flow velocity (surface?)*

Response: The depth-averaged mean flow velocity values were derived from flow measurements conducted during the calibration measurements using following systems: an magnetic-inductive flow meter OTT MF Pro (at the Albula and Navisence sites), a radar-based stage sensor Vegapuls WL 61 (Avançon de Nant site), and a 2-D laser sensor TiM551 by SICK AG© (Erlenbach site).We have added to the legend the name of the different devices used to derive flow velocities (Lines 145-147).

**Comment 7:** *Line 146: you mean "uniform mixture"?*

Response: This is indeed what is meant here. Our experience from a recent publication (Nicollier et al., 2022) has shown that people tend to be quite confused by the combination of "uniform" and "mixture". Therefore, in order to avoid any confusion, we have decided to formulate this sentence letting aside the word "mixture" ("with a fixed number of grains for each of the ten particle-size").

- Nicollier, T., Antoniazza, G., Rickenmann, D., Hartlieb, A., and Kirchner, J.W.: Improving the calibration of impact plate bedload monitoring systems by filtering out acoustic signals from extraneous particle impacts. *Earth Space Sci.*, *9*, e2021EA001962, https://doi.org/10.1029/2021EA001962, 2022.

**Comment 8:** *Line 173-174: what happens when several grains hit the plate simultaneously? (the question concerns SPG in the field)*

Response: In the current signal processing procedure and packet definition, such a situation might result in a large packet containing multiple larger peaks. Such a packet would be a typical example of signal saturation. We address this issue in further detail in Section 4.4 in order to not overload the methodology part with too much information. Please refer to our response to your comment no. 30.

**Comment 9:** *Table 3: it is very difficult to understand this table and its title*

Response: We have reformulated the caption of the table and have removed the information concerning the manual sorting of the upper classes, which did not add any relevant information and might have been confusing.

**Comment 10:** *Line 180: Is this equation is site specific? First the material: can we consider that all sensors have exactly the same response? Secondly the bedload data may be specific to Erlenbach (mode of transport, grains velocity, density…)?*

Response:     This is an interesting question. In fact, this equation has been implemented at several sites, regardless of the different factors you have listed. From calibration measurements at different field sites (Rickenmann et al., 2014; Wyss et al. 2016a), from flume experiments using particles from different field sites (Wyss et al. 2016b), and from controlled impact tests conducted at multiple sites (Antoniazza et al., 2020), we know that SPG plates have in general a comparable signal response. But if we combine the small variability in signal response among plates to the variability in transport conditions and bedload specificities, it would be reasonable to expect a certain variability in the instrument response. In the discussion (Lines 536-612), we cite the flow velocity, the transport mode, the saltation length, etc. as possible factors influencing the signal response, whether it is the amplitude of the geophone signal or the number of detections per unit mass. However, these factors become particularly relevant when it comes to the transfer of a calibration relationship to another monitoring station. As long as we consider a given station per se, we consider that the effects of these factors are included in the site-specific calibration relationship.

- Antoniazza, G., Nicollier, T., Wyss, C. R., Boss, S., and Rickenmann, D.: Bedload transport monitoring in alpine rivers: Variability in Swiss plate geophone response, *Sensors*, *20*, https://doi.org/10.3390/s20154089, 2020.

- Rickenmann, D., Turowski, J. M., Fritschi, B., Wyss, C., Laronne J.B., Barzilai, R., et al.: Bedload transport measurements with impact plate geophones: comparison of sensor calibration in different gravel-bed streams, *Earth Surf. Processes Landforms*, *39*, 928–942, https://doi.org/10.1002/esp.3499, 2014.

- Wyss, C. R., Rickenmann, D., Fritschi, B., Turowski, J., Weitbrecht, V., and Boes, R.: Measuring bed load transport rates by grain-size fraction using the Swiss plate geophone signal at the Erlenbach, *J. Hydraul. Eng.*, *142*(5), https://doi.org/10.1061/(ASCE)HY.1943-7900.0001090,04016003, 2016a.

- Wyss, C. R., Rickenmann, D., Fritschi, B., Turowski, J., Weitbrecht, V., and Boes, R.: Laboratory flume experiments with the Swiss plate geophone bed load monitoring system: 1. Impulse counts and particle size identification, *Water Resour. Res.*, *52*, 7744–7759, https://doi.org/10.1002/2015WR018555, 2016b.

**Comment 11:** *Line 183: I don't understand. You use the measured packets with Eq.1 for computing each size class present in a bedload mixture?*

Response:     Eq. 1 relates the amplitude of the SPG signal to the size of an impacting particle. So yes indeed, the AH method proposed by Wyss et al. (2016a) relies on this relationship to estimate the particle sizes present in a bedload mixture. In order to clarify this, we have rephrased the sentence on Lines 200-201.

- Wyss, C. R., Rickenmann, D., Fritschi, B., Turowski, J., Weitbrecht, V., and Boes, R.: Measuring bed load transport rates by grain-size fraction using the Swiss plate geophone signal at the Erlenbach, *J. Hydraul. Eng.*, *142*(5), https://doi.org/10.1061/(ASCE)HY.1943-7900.0001090,04016003, 2016a.

**Comment 12:** *Line 200: why "in the lower size class"? I would expect that only large particles produce extraneous impacts?*

Response:     It is right that only large particles produce impacts that are detectable by the SPG system (e.g. Nicollier et al. 2022). However, when the SPG system is impacted by such a large particle, the propagating signal attenuates along its travel path, and the neighboring sensors only detect a fraction of the energy released on the impacted plate. The apparent packets recorded by these sensors will therefore mainly be "falsely" classified in lower size classes. We have rephrased the sentence on Lines 209-211 to clarify this.

- Nicollier, T., Antoniazza, G., Rickenmann, D., Hartlieb, A., and Kirchner, J.W.: Improving the calibration of impact plate bedload monitoring systems by filtering out acoustic signals from extraneous particle impacts. *Earth Space Sci.*, *9*, e2021EA001962, https://doi.org/10.1029/2021EA001962, 2022.

**Comment 13:** *Line 234-235: "The transported bedload mass associated with an individual signal packet is strongly dependent on the size of the impacting particle" what is difficult with such a sentence is that we don't really understand if you describe the movement of a single particle or of a bedload mixture.*

Response:     We agree that beginning the sentence with "bedload mass" can be somewhat misleading. We have rephrased the sentence and have replaced "transported bedload mass" with "particle mass" to make it clearer that we describe a single particle (Line 253).

**Comment 14:** *Line 237: hard to follow. If I understood well you will apply a threshold to both amplitude and frequency. In the next sentence "lower threshold" and "upper threshold" concerns amplitude or frequency?*

Response:     The lower threshold is based on amplitude only, while the upper threshold is based on both amplitude and frequency information. We have reformulated several parts of Section 2.5.2 (Lines 252-303) in order to clarify the origin and the aim of these thresholds.

**Comment 15:** *Line 250: could you tell a bit more about these coefficients?*

Response:     In order to avoid overloading this already quite complex section, we have decided to add only a bit of additional information on this topic on Lines 270-271 to clarify the origin of these coefficients. More details can then be found in the indicated reference (Nicollier et al., 2022).

- Nicollier, T., Antoniazza, G., Rickenmann, D., Hartlieb, A., and Kirchner, J.W.: Improving the calibration of impact plate bedload monitoring systems by filtering out acoustic signals from extraneous particle impacts. *Earth Space Sci.*, *9*, e2021EA001962, https://doi.org/10.1029/2021EA001962, 2022.

**Comment 16:** *Line 255: where do these equations come from? your experiments?*

Response: Yes, the entire Section 2.5.2 is based on the single-grain-size experiments we have run at the Obernach facility. To underline this, we have reminded on Lines 277-278 as well as in the caption of Figure 5 that individual grain-sizes were fed into the flume.

**Comment 17:** *Line 257: If I understood, by replacing Dmj (the sieve sizes) in Eq 4 and 5 the objective is to isolate the packets associated with a given size class? Not clear (same for figure 5)*

Response: Yes, by replacing the lower and upper sieve sizes of each grain-size class fed into the flume, we are able to derive the lower and upper thresholds used to assign a packet to a given class *j*. We have clarified this information on Lines 277-278, 281-283, and 298-299.

**Comment 18:** *Line 279: you must imagine that you present to somebody who knows nothing about your work. Since I am reading, I am still lost with your upper and lower threshold.*

Response: We agree that Section 2.5.2, in which the thresholds are being introduced, was perhaps not straightforward enough to follow. We have considerably reworked this section, so that the definition of thresholds should now be clearer to follow.

**Comment 19:** *Line 282: YES!! I have my answer!!*

Response: With the changes made to Section 2.5.2 we hope to have clarified this point already before the reader reaches Section 2.5.3.

**Comment 20:** *Line 284: The link between Eq.6 and 7 is not clear (I suppose that med station refers to all samples i)*

Response: We have clarified that the median is computed over all samples *i* on Line 314.

**Comment 21:** *Line 287: providing a general methodology for reducing the measurement uncertainties in a given site is already a nice objective. But the passage to a general inter-site calibration term is not trivial. It supposes that beside the plate response, all sites share the same transport characteristics. For instance if grains saltate over long distances (and different station length form one river to another) can we be sure that the impact rate reflect the real transport?*

Response: You are pointing to an important aspect. The goal of this paper is to investigate the feasibility of such a general calibration procedure, mainly by focusing on the bias introduced by the detection of apparent packets. The fact that both total and fractional fluxes are generally overestimated at low transport rates and underestimated at high transport rates (see Section 4.4) suggests that, already at a given site, variations of the transport conditions affect the SPG signal

response. When considering site-to-site variations of the transport conditions, it would therefore be reasonable to expect even larger differences. This is well visible when comparing the estimates obtained for the Erlenbach with the results obtained for the other three streams. The stochastic nature of bedload transport makes it difficult to establish accurate relationships between the transport mode, the transport rate and the SPG signal response. We have recently started an uncertainty analysis of the site-specific as well as of the general calibration relationships in order to better understand the relevance of such factors. But it is motivating that although there are multiple factors that are not yet integrated in the approach, the global calibration coefficient is working reasonably well.

**Comment 22:** *Line 346: the scatter is lower? Not so clear in the figure*

Response:  We agree that for the largest four classes, there is barely a difference in scatter between the AH and the AF method. However an important reduction of scatter of the $k_{b,i,j}$ coefficients is well visible for the six smallest classes. We have modified the sentence on Lines 378-379 accordingly.

**Comment 23:** *Figure 9: add the light grey dots in the legend*

Response:  The legend has been changed as suggested. We  also modified the legend of Figure S2 in the supporting information accordingly.

**Comment 24:** *Line 407: Direct sampling depends on the mesh size and the SPG measurements concern sizes >12mm. We know that in many mountain streams the contribution of gravels and sand can be very large. How can you take this into account?*

Response:  You are right. As mentioned earlier, due to the lower detection limit of the SPG system and in order to reduce the flow resistance of the net sampler, we used a mesh size of 8 mm at the Albula and the Navisence, and a mesh size of 6 mm at the Avançon de Nant site, where the flow was much weaker. The SPG systems also does not detect particles < ~10 mm. As such, there is still a range of bedload particle sizes that may not be detected. An interesting indication about the transported bedload mass that is missed by the SPG system, can be found in the bedload samples collected at the Avançon de Nant site, where we used the net with smallest mesh size across all calibration campaigns (6 mm). Due to clogging of the net particles down to 4 mm have also been trapped, and later sieved and weighted. On average over all samples collected at the Avançon de Nant, the mass of particles ranging from 4 to 9.5 mm (and thus not detected by the SPG system) represented proportions of the total sampled mass of around 0.16. Proportions of up to 0.44 have been observed at the Avançon de Nant. Rickenmann et al. (2018), report mean proportions of particles with sizes 2 mm<D<10 mm of around 0.22 from two other sites equipped with the

SPG system (Fischbach and Ruetz). This underlines that the SPG system only detects a part of the whole bandwidth of the transported bedload particle sizes.

Regarding the sensing, a possible solution could be to combine the SPG system to other types of sensors that have a higher sensitivity to smaller fractions. However, this could be quite challenging in such turbulent streams (e.g. ADCP). The use of accelerometers additionally to geophone sensors and thinner steel plates might already help to decrease the lower detection threshold (as indicated by both geophone and accelerometer sensors used at the Albula field site, see Rickenmann et al., 2017). A different approach would be to extrapolate the obtained grain-size distributions towards smaller fractions using models (e.g. Schneider et al., 2016) fed with the morphological and flow characteristics of the investigated site.

- Rickenmann, D., Antoniazza, G., Wyss, C.R., Fritschi, B. and Boss, S.: Bedload transport monitoring with acoustic sensors in the Swiss Albula mountain river. *Proceedings of the International Association of Hydrological Sciences*, *375*, 5–10, https://doi.org/10.5194/piahs-375-5-2017, 2017.

- Rickenmann, D., Steeb, N., and Badoux, A.: Improving bedload transport determination by grain-size fraction using the Swiss plate geophone recordings at the Erlenbach stream, in River Flow 2018, *Proceedings of the 9th Int. Conference on Fluvial Hydraulics,* 8 pp., https://doi.org/10.1051/e3sconf/20184002009, 2018.

- Schneider, J. M., Rickenmann, D., Turowski, J. M., Schmid, B., and Kirchner, J. W.: Bed load transport in a very steep mountain stream (Riedbach, Switzerland): Measurement and prediction, *Water Resour. Res.*, *52*, 9522–9541, https://doi.org/10.1002/2016WR019308, 2016.

**Comment 25:** *Figure 10: How do you explain a small tendency to overprediction for lower transport?*

Response: Inversely to the underestimated transport rate at high transport intensities (Section 4.4), we can expect a stronger signal response (i.e. $k_{b,i,j} > k_{b,med,j}$) at lower transport intensities due to lower flow velocities, shorter saltation lengths, and unsaturated signal, which would all result in an increased amount of detected packets per unit weight. We have added this consideration on Lines 578.

**Comment 26:** *Line 405: it questions on the pertinence of a general calibration coefficient. Also, many sites are equipped with SPG. Could it be possible to test the calibration coefficient with other sites?*

Response: The presented procedure uses frequency information and thus relies on the recording of parts of the raw signal at the field monitoring station. Until now, packets and the signal they contain are being recorded at the four stations presented in this study only. At the other sites equipped with the SPG system, the recording of packets has unfortunately not been implemented yet. There, the field computers record only summary values such as the number of

impulses per minute, maximum amplitude, etc. which do not allow any frequency-based analysis of the SPG signal. Since the three natural sites show a relatively good agreement, it would indeed have been of major interest to apply the developed procedure to other calibrated field monitoring stations. A sentence has been added on Lines 119-120 to explain the choice of these four stations.

**Comment 27:** *§4.1: the paper is already very long and not easy to read. Is this paragraph really necessary? Or maybe to be reduced.*

Response:     We have shortened the section as suggested.

**Comment 28:** *Line 484: It partly answer to see my previous comment about limitations of a general calibration*

Response:     For further explications, please refer to our response to your comment no. 26.

**Comment 29:** *Line 488: Huge question which also concerns the contribution of finer fraction, how the saltation length of large elements affect the SPG detection…*

Response:     Yes, indeed. In our opinion this is one of the most important sections, that aims to remind to the reader that the development of a general calibration procedure certainly requires a large set of calibration measurements, but above all a clean and consistent bedload sampling across all sites, which is challenging to evaluate. As mentioned on Lines 536-557, keys to improve the calibration accuracy are certainly to collect large samples and avoid short sampling intervals. Doing so enables us to 1) average over stochastic factors such as the impact location, which depends on the saltation length, and 2) to sample a representative range of particle sizes. Considering these points, it is not necessarily a disadvantage that the SPG system does not detect particles smaller than around 10 mm. In fact, if smaller fractions (< 9.5 mm) had to be sampled accurately, the whole sampling procedure would be significantly more difficult due to the use of a smaller mesh size (i.e. higher flow resistance, faster filling, shorter sampling intervals).

**Comment 30:** *Line 519: "In our SPG data, we have observed long packets containing multiple large peaks corresponding to several impacts occurring so quickly after one another that they were not detected as separate packets". It's a shame that this comment appears at the end of the manuscript because it's the image we immediately have in mind, which doesn't match the definition of the package in Figure1. It could be worth to explain early how you considers this aspect in your analysis.*

Response:     Following your suggestion, we have added a sentence on Lines 227-229 to address this problematic already earlier in the text. The evaluation of the relevance of packets containing multiple impacts and the development of a

procedure to split them into sub-packets could represent an interesting topic for future research.

**Comment 31:** *§4.5 same comment. From the beginning I suspect grains velocity to play a role in the SPG response. I regret that this parameter is totally occulted in the paper. Even if you do not consider it in the analysis, it could be introduced earlier.*

Response: The lack of continuous flow stage recordings is certainly one of the missing links in this study. It would have been interesting to combine the findings from the flume experiments performed by Wyss et al. (2016c) to a new and extensive set of stream flow measurements from four different sites to improve the presented procedure. But the fact that the three watercourses characterized by a natural bed show similar instrument responses across multiple grain-size classes even though they are characterized by different mean flow velocities (Figure 8), possibly suggests that further factors such as the bed morphology also heavily influence the SPG response.

In order to inform the reader already earlier in the text about the relevance of the flow velocity, we have added some explanations on Lines 123-127.

- Wyss, C. R., Rickenmann, D., Fritschi, B., Turowski, J., Weitbrecht, V., Travaglini E, et al.: Laboratory flume experiments with the Swiss plate geophone bed load monitoring system: 2. Application to field sites with direct bed load samples, *Water Resour. Res.*, *52*, 7760–7778, https://doi.org/10.1002/2016WR019283, 2016c.

**Further minor changes**

We have also made some further minor changes to the original manuscript. These mainly concern typos, update of recently published references, and general reformulations of terms or sentences. All changes can be found in the "tracked-changes" version of the manuscript.

---

## Author Comment (AC2)

**Response to the comments made by Referee #2**

Dear Dan Cadol,

We appreciate your valuable comments and interesting ideas regarding some key elements of the manuscript, which have significantly helped to improve it in our opinion. We agreed with most of your suggestions, and have made the modifications accordingly. Below, the comments are reported in italics, and our responses in normal font (blue color). The indicated line numbers refer to the tracked-changes version of the revised manuscript.

**Comment 1:** *This paper presents a valuable synthesis of efforts to clean SPG signals and make comparisons across field sites. While complex, I found the 'apparent' impact cleaning method to be clear in the end. But key to my understanding was Figure 7. I was a bit lost regarding the thresholds until I came to that valuable figure. Even after cleaning, the bedload flux prediction errors up to 5-fold suggest that there is still work to be done. But this is a major step forward. And it's difficult to know how much of the discrepancy is due to direct bedload measurment uncertainty or the SPG signal features and processing.*

*My main question/suggestion is also related to Figure 7. Using fixed thresholds is reasonable due to its simplicity. But Fig 7b suggests that thresholds that are functions of both MaxAmp and MaxAmp/f would exclude apparent pulses and retain real pulses more reliably, without the step-like effect of the fixed value thresholds mentioned by the authors. Given the computational and field effort required to obtain the MaxAmp and f_centroid data, this extra bit of analysis seems trivial by comparison. But perhaps the need to automate the signal reduction makes such an approach untenable? Or the exact slope and intercept of the dividing lines in Fig 7b that I am suggesting varies from site to site? A brief exploration of the differences in the MaxAmp vs. MaxAmp/f plots for different sites and flume setups, or just a comment on the differences, might help.*

*Another possible advantage of sloped thresholds is that you could make them nonoverlapping, eliminating the double counting of packets. I wasn't entirely convinced by the statement that double counting impacts is a non-issue.*

Response:     We kindly thank you for your positive comments.

We have considerably reworked Section 2.5.2 (Lines 252-303) to make the threshold definition clearer.

Your idea of thresholds defined by a smooth function rather than a step function is interesting. In a recent study, we showed that the signal response of the SPG system was similar at all four sites (Figure 7, Nicollier et al. 2022). Using a unique function to split real from apparent packets would thus be

possible, even though we expect some small variations between sites. However, what is certain is that defining the exact slope and intercept of the site-specific dividing lines between real and apparent packets is impossible because the true origin of a packet in field data cannot be determined. The only dataset that enables us to do so is the one resulting from the flume experiments at Obernach conducted with the partition wall. In the study mentioned earlier (Nicollier et al. 2022), we did not select real packets using class thresholds but simply on the basis of two criteria, one of them similar to Eq. 3. This procedure is therefore comparable to your suggestion of a continuous function. Before developing the concept of lower and upper thresholds, we tested whether combining such a simple criterion as the amplitude threshold defined by Wyss et al. (2016a) would result in satisfying transport and grain-size estimates over all four stations (Table 3). Even though the results were close to the estimates obtained with the AF method, they were never as good.

In our opinion the overlapping class boundaries are convincing for the following reasons:

(i) We used the 75[th] percentile values of the ranges visible in Fig 5 to define the thresholds. Most of the impacts will therefore generate packets that are being classified in classes that are lower than the class $j$ corresponding to the true grain size. Overlapping class boundaries therefore enables a less strict classification of the few packets that are on the edges of the classes. In Figure 7b out of 2256 packets recorded by G2 (blue), 144 packets have been counted twice. But interestingly, not a single of the 153 packets recorded by G1 (red) encompassed by the class boundaries has been counted twice. We could therefore state, even if this might be somehow exaggerated, that the overlapping class boundaries also contribute to an increase of the signal to noise ratio.

(ii) Counting packets twice should not introduce any type of bias because in the second step of the presented signal conversion procedure, we have calibrated the SPG system by dividing the number of packets recorded in a given class by the bedload mass of the corresponding grain-size class. The packets counted twice are therefore included in the calibration coefficients $k_{b,j}$, resulting in will slightly larger values.

We have added these considerations on Lines 519-522.

- Wyss, C. R., Rickenmann, D., Fritschi, B., Turowski, J., Weitbrecht, V., and Boes, R.: Measuring bed load transport rates by grain-size fraction using the Swiss plate geophone signal at the Erlenbach, *J. Hydraul. Eng.*, *142*(5), https://doi.org/10.1061/(ASCE)HY.1943-7900.0001090,04016003, 2016a.

- Nicollier, T., Antoniazza, G., Rickenmann, D., Hartlieb, A., and Kirchner, J.W.: Improving the calibration of impact plate bedload monitoring systems by filtering out

acoustic signals from extraneous particle impacts. *Earth Space Sci.*, *9*, e2021EA001962, https://doi.org/10.1029/2021EA001962, 2022.

**Comment 2:** *Fig 8b. There are fewer pulses/kg of the two smallest particle classes relative to the third smallest. Is this decline due to saltation? It's an interesting result, which was masked by the counting of apparent packets in the amplitude-only thresholding method. I would appreciate some thoughts about it in the discussion.*

Response: Thank you for this good observation. A similar decrease towards the two smallest classes has already been described by Wyss et al. (2016b). On the basis of flume experiments they could show that the number of impulses detected per unit weight strongly decreases towards zero after reaching a peak value at around 25-37.5 mm. Because of the steel plate's mass, the low energy released by impacts of such small particles but certainly also because of the longer saltation lengths, the detectability of particles decreases with decreasing particle size for the two lowest size classes. We have followed your suggestion and have added some comments on this interesting characteristic of the calibration coefficients on Lines 522-525.

- Wyss, C. R., Rickenmann, D., Fritschi, B., Turowski, J., Weitbrecht, V., and Boes, R.: Laboratory flume experiments with the Swiss plate geophone bed load monitoring system: 1. Impulse counts and particle size identification, *Water Resour. Res.*, *52*, 7744–7759, https://doi.org/10.1002/2015WR018555, 2016b.

**Comment 3:** *Line 545: Why are there fewer impacts/kg when the particle (or flow) velocity is higher? I would think greater particle velocity would produce more readable impacts, and thus more impacts/kg? I think the text is suggesting that it's because of more saltation, and thus skipping over the plate. Is this correct? Just a little more clarification of your hypotheses for this feature in the data would be appreciated.*

Response: Yes, this is correct. Wyss et al. (2016b) had already proposed that this is due to fast moving particles being less likely to collide against the Swiss plate geophone than slower moving ones, which are more frequently in contact with the bed. A secondary effect concerns is the increased energy released by an impact with higher flow velocities. Wyss et al. (2016b) hypothesizes that smaller particles that are transported faster collide with more energy against the Swiss plate geophone than slower particles, probably due to increasing saltation height or increasing turbulence with increasing $V_f$. Still, the better detectability of particles that arises from the higher impact energy seems to be insufficient to compensate the strong reduction of the number of impacts on a plate with increasing flow velocities. This possibly arises from the fact that larger flow velocities (without increased turbulence) may also lead to flatter saltation trajectories, thus decreasing the vertical component of the impact force. We have tried to clarify these effects of the flow velocity on Lines 593-594 and Lines 598-601.

**Comment 4:** *Line 590: It's good to make clear that uncertainty in the direct measurements used for calibration is very real, and that this may contribute to weaknesses or biases in the predictive power of the modeled estimate. You do mention this, I just think it can be easily forgotten in general, and perhaps merits emphasis.*

Response: We agree that this point is essential and can only be stressed more. We have added a sentence that underlines this problematic on lines 555-556.

**Further minor changes**

We have also made some further minor changes to the original manuscript. These mainly concern typos, update of recently published references, and general reformulations of terms or sentences. All changes can be found in the "tracked-changes" version of the manuscript.

---

## Author Comment (AC3)

**Response to the comments made by Referee #3**

Dear Roger Kuhnle,

We appreciate your valuable comments regarding some key elements of the manuscript and thank you for having raised the interesting issue of real packets being potentially eliminated. We believe that your comments, questions and suggestions significantly helped to improve the manuscript. We agreed with most of your suggestions, and have made the modifications accordingly. Below, the comments are reported in italics, and our responses in normal font (blue color). The indicated line numbers refer to the tracked-changes version of the revised manuscript.

**Comment 1:** *The manuscript reports on a new method to calibrate the Swiss plate geophone (SPG) which uses a combination of data collected in a laboratory flume and at four different field sites. The SPG has been shown in previous studies to be an excellent indirect method to measure the rate and size of bed load transport in gravel bedded streams and rivers. This study further develops the science of turning impact data into quantitative data of the transport of bedload. The study of bed load transport using impact plates at several field sites has indicated that a general calibration of the SPG has been difficult to develop for a variety of reasons. This study uses a combination of amplitude and frequency to calibrate the SPG in the quest for a general calibration relation to turn impact data into quantitative values of mass and grain size for gravel bed load transport. This manuscript contains much valuable information and should be published, however, suggestions for improvement of the presentation are given below.*

Response: Thank you for these positive comments on our work.

**Comment 2:** *Lines 246-251: In these lines it is related how all packets were filtered using equation (3) and packets which do not meet this criterion are ignored in further analysis. It is clear to me how and why this was done, however, in Figure 7 there appears to be substantial overlap between real (blue) and apparent (red) peaks measured in the flume experiments. How many real peaks were rejected using this criterion in the flume experiment data? Also, could the authors estimate how many real peaks were rejected from the 4 field data sets considered in the study? I believe text should be added to the manuscript discussing this issue.*

Response: Thanks for this interesting question that is not trivial to answer. We have performed single-grain size experiments in two different ways. The first type, which is also the most relevant one for this study, was conducted without the partition wall. Eq. 3 has been applied to the data collected during these experiments only. A comparison of the signal responses before and after the application of the criterion described in Eq. 3 is shown in Figure 5. After the application of the criterion about 39% of all packets are remaining. This

information has been added at the end of section 2.5.2, on Line 290. These are being used to derive the lower and upper threshold values. In absence of the partition wall, it is difficult to give any indication about the number of real packets that were filtered out by Eq. 3, because the true location of an impact at the origin of a packet cannot be verified. This is also true for the field measurements.

The following short quantitative analysis of Figure 7 could help the reader to better understand the effect of the application of the two-dimensional thresholds on the number of real and apparent packets. As mentioned on Lines 167-169, the single grain-size experiments performed in presence of the partition wall are used to illustrate the performance of the two methods (see Figure 7). Eq. 3 has therefore not been directly applied to the dataset resulting from these experiments, only the class threshold values mentioned earlier have been applied. Through the application of the AF thresholds (Fig. 7b and d), an important part of the packets detected by G1 and G2 is being eliminated. "The AH thresholds encompass in total 1945 packets for the shielded geophone G1, and 4823 packets for the unshielded geophone plate G2. In comparison, the AF thresholds encompass in total 159 packets for the shielded geophone G1, and 2202 packets for the unshielded geophone plate G2 (counting the packets in the overlapping class boundaries only once)." (inserted in Lines 356-359). An important point is that the blue dots (packets detected by the unshielded sensor G2) that are in the overlapping area with red dots (packets detected by the unshielded sensor G1), are most probably not real packets. We have added a short explanation to clarify the reason for this certainly confusing but very important point on Lines 345-347. The overlapping area arises from the fact that a seismic wave generated by an impact on the concrete bed follows a similar path towards both sensors, resulting in the recording of two apparent packets with comparable characteristics. We expect that also packets originating from "unclean" impacts, for example close to the edge of a steel plate, or with a large horizontal component (low amplitude), could be found in this overlapping area, but unfortunately we cannot indicate how many. The fact that there is, most probably, not a sharp line splitting real from apparent packets, might be a good explanation for why the most optimal class thresholds include a part of the packets located in this overlapping area.

**Comment 3:**  *Lines 239-240, and 257-260: In these sentences the lower and upper thresholds for the amplitude-frequency method are described.  Is it correct that the lower threshold (V) was based on the minimum grain size of the size fraction and the upper threshold (V Hz) was based on the maximum grain size of the size fraction being considered?   The clarity of the text should be improved to make it easier for the reader to interpret the details of how this technique was implemented.*

Response: Yes, this is correct. We have reformulated several sentences in Section 2.5.2 (Lines 252-303) and hope that it clarifies the entire process leading to the lower and upper thresholds.

**Comment 4:** *Figure 9: This Figure is too small and has too much information contained in it. This renders this Figure very difficult to interpret other than for a general impression of the data trends. Consider simplifying this Figure or possibly presenting this information on two Figures.*

Response: We have modified the format of Figure 9 as well as the corresponding Figure S1 in the Supporting Information in order to improve their readability. However, the main goal of this figure is really to give a qualitative impression of the accuracy of the transport rate estimates and is not meant as standalone element. All the values required to evaluate the performance of the two methods are listed just below the Figure in Table 5.

**Comment 5:** *Lines 394-400, Table 5: The comparison of the two methods for arriving at quantitative rates and sizes of bed load is interesting. Was the criterion in eq (3) applied to the data before the amplitude histogram (AH) method was implemented? It is clear that the criterion it eq (3) was used as part of the technique for the amplitude-frequency (AF) method. Some text should be added to make it clear as to whether eq (3) was applied in relation to the data before applying the AH method.*

Response: Thank you for this important remark. In order to compare the new AF method with the original AH method developed by Wyss et al. (2016a), we did not apply any filtering of packets using Eq. 3 before implementing the AH method, since this was not part of the original procedure. We have added this information on Lines 273-274.

- Wyss, C. R., Rickenmann, D., Fritschi, B., Turowski, J., Weitbrecht, V., and Boes, R.: Measuring bed load transport rates by grain-size fraction using the Swiss plate geophone signal at the Erlenbach, *J. Hydraul. Eng.*, *142*(5), https://doi.org/10.1061/(ASCE)HY.1943-7900.0001090,04016003, 2016a.

**Comment 6:** *Lines 571-598: It is clear that the AF method performed better than the AH method in some cases such as the Erlenbach, however, it is also clear that for the other 3 field sites the AH method yielded results for bed load that were quite close to that obtained from the physical samples. It is also not clear whether the general calibration calculated in this study would give.*

Response: We agree that the results obtained with the AH method for the three "natural" sites were already good and of similar accuracy. Indeed, all three sites had already very similar site-specific calibration relationships before filtering out apparent packets (see Fig. 8a). The main motivation behind the AF method was not necessarily to improve the accuracy of site-specific calibration relationships but rather to better understand and possibly reduce large

differences among (all) sites. Considering Fig. 8b, one can notice that through applying the AF method, the differences between the calibration relationships of the three natural sites and the ones of the Erlenbach site have been reduced. Most results in the present study seem to suggest that the improvements arising from the use of the AF method mainly concern the comparison of the Erlenbach and the other datasets. However, as one can see in Fig. 8, also the $k_{b,j}$ coefficients of the three other sites underwent important changes (see also Table S4). In our opinion, the fact that the accuracy of the estimates at the three natural sites has not been reduced with regard to the AH method, even after the removal of an important part of the detected packets, supports a use of the AF method. This considerations has been added on Lines 493-494.

While the lack of accurate flow velocity measurements is certainly one of the critical points of the study, one could argue that another one is the low variability between the site-specific calibration relationships already before implementing the AF method. Indeed, it would have been extremely interesting to test the method on a larger number (and variety) of sites. Unfortunately, these are the only sites at which a full geophone signal has been recorded during calibration measurements. These considerations have been added on Lines 504-508.

**Further minor changes**

We have also made some further minor changes to the original manuscript. These mainly concern typos, update of recently published references, and general reformulations of terms or sentences. All changes can be found in the "tracked-changes" version of the manuscript.

---

## Author Response (AR2)

**Response to the comments made by the Associate Editor**

Dear Claire Masteller,

We appreciate your valuable comments regarding the fluidity of the text and the several suggestions you have made to improve the readability of figures. Your comments, questions and suggestions were helpful to improve the manuscript. We agree with most of your suggestions, and have made the modifications accordingly. Below, your comments are reported in italics, and our responses in normal font (blue color). The indicated line numbers refer to the tracked-changes version of the revised manuscript including our new modifications.

**Comment 1:** *Dear authors,*
*Thank you for your submission to E-Surf. Three reviews of the manuscript were generally positive and agreed that the results represent a novel and important contribution towards a generalized calibration for measurements of bedload flux via the Swiss Plate Geophone system. The reviewers asked for some clarifications throughout the manuscript that I feel have been sufficiently addressed - thank you for your thoughtful engagement with reviewer comments.*
*I have gone through the revised manuscript and am suggesting some minor edits for clarity and in order to streamline some sections of the manuscript. I have also made a number of suggestions on figure design to improve readability and clarity. Please find these suggestions in the attached PDF. ecause these are mainly comments on the text, not on the methods or analysis, I am marking these as minor revisions.*

*All the best,*
*Claire Masteller*

Response: We are very happy to learn that the concerns raised by the three reviewers were sufficiently addressed during the first revision stage. We agree that the clarity and fluidity of several sections could however still be improved, especially in Section 3.3 (starting on Line 410). We are also very grateful for your excellent suggestions on how to improve the figures and definitely recognize that the choice of the greyish background color in multiple figures was not as ideal as we initially thought. Thank you!

**Comment 2:** *Line 15: "towards the development of"*

Response**:** We have rephrased the sentence as suggested (see Line 15).

**Comment 3:** *Line 16: Can this be replaced with "channels"? - i recognize that what you've used is a more generic phrase, but also may be a bit abstract*

**Response:** We agree with you and have changed the wording accordingly (see Line 16).

**Comment 4:** *Line 17: This is a bit vague - consider revising towards a more specific statement such as "bedload mass flux" or "the intensity and characteristics of transported bedload"*

**Response:** We have rephrased the sentence following your second suggestion (see Lines 17-18).

**Comment 5:** *Line 20: "outside of"*

**Response:** Thank you for having spotted this oversight. We have added the missing word "of" (Line 20).

**Comment 6:** *Line 20: Deleted word "here"*

**Response:** We have deleted the word "here" (Line 21).

**Comment 7:** *Line 21: Second use of calibration in this sentence - may consider revising to "direct field measurements"*

**Response:** The sentence has been modified as suggested (Line 22).

**Comment 8:** *Line 28: Deleted comma*

**Response:** Modified as suggested (Line 29).

**Comment 9:** *Line 31: "including"*

**Response:** Modified as suggested (Line 32).

**Comment 10:** *Lines 44-45: sort of awkward phrasing - maybe "errors spanning multiple orders of magnitude"? "errors on the scale of multiple orders of magnitude"?*

**Response:** The sentence has been rephrased following your second suggestion (Lines 46-47).

**Comment 11:** *Line 46: But is this ultimately true for SPG? the measurements are ultimately fairly concentrated whereas something like seismic monitoring would integrate measurements from wider areas*

Response: Correct. With regard to seismometers, the SPG system has a relatively limited coverage. This statement, however, specifically refers to the integration of bedload transport over entire river cross-sections, and in that sense we think that the SPG system represents an important improvement in term of coverage as compared to direct sampling techniques. To be more precise, we have replaced the term "large spatial coverage of river transects" by "complete coverage of selected river transects" (see Line 49)

**Comment 12:** *Lines 57-58: I suggest adding relevant citations for each of the applications listed here where possible- many from this group!*

Response: We have added multiple citations to each application and have attempted to include various research groups (see Lines 60-65).

**Comment 13:** *Line 85: I suggest replacing this with laboratory or eliminating the word controlled all together - I think the fact that they are controlled is reasonably implied.*

Response: We agree that the word "controlled" is not necessary and have eliminated it throughout the manuscript (see Lines 19, 75, 79, 90, 157, 158, 511 and 679). At some locations we have replaced the word "controlled" with the word "flume".

**Comment 14:** *Line 97: "with a minimum diameter of 10 mm"*

Response: Modified as suggested (see Line 102).

**Comment 15:** *Lines 100-101: Deleted "the outdoor"*

Response: Removed as suggested (see Line 106).

**Comment 16:** *Line 101-102: Replacing "ranging from a few seconds to one hour" by "for the full duration of each measurement event, ranging in duration from a few seconds to one hour."*

Response: Rephrased as suggested (see Lines 105-107)

**Comment 17:** *Line 102-106: This statement seems a bit out of place - The last sentence of this paragraph that was removed put this statement into context more clearly. I am not sure if this is ultimately necessary? Consider revising*

**Response:** We agree that this statement is not necessary to understand the content of this study. Since the manuscript is already quite long, we have decided to remove these lines (see Lines 107-111).

**Comment 18:** *Line 104: "rather"*

**Response:** Please refer to the response to your previous comment.

**Comment 19:** *Figure 1: I suggest that you add text labels next to one and two that say "uniaxial geophone sensor" and "elastomer element" for clarity*

*in B, i find the numeric axes labels to be smal and ahrd to see and the lines representing the therhsold amplitudes hard to see. I might recommend changing the aspect ratio of this figure so the panels are vertical so B can be larger? an increase in line weight would also help*

**Response:** We have followed all your suggestions related to Figure 1 and we think that the content of the new figure is now easier to read (see line 112).

**Comment 20:** *Line 115: replace with field?*

**Response:** We have followed your suggestion since the word calibration already appears in the title as well as in the following sentence (see Line 120).

**Comment 21:** *Line 119: channel morphology*

**Response:** We have added the word "morphology" as suggested (see Line 124).

**Comment 22:** *Line 120: Deleted "full"*

**Response:** This sentence has been removed since similar information was already given on Line 106.

**Comment 23:** *Line 120: Deleted "carried out"*

**Response:** Deleted as suggested (see Line 125).

**Comment 24:** *Figure 2: For non-swiss readers, it may be helpfult to include a general map of these locations*

**Response:** This is a good idea, thanks! We have modified the figure and the caption accordingly (see Lines 137-144).

**Comment 25:** *Line136-137: a) also has a crane-mounted sampler*

Response: We have rephrased the caption to avoid any confusion (see Lines 141-144).

**Comment 26:** *Line 151: Deleted "Controlled"*

Response: Deleted as suggested (see Line 157).

**Comment 27:** *Line 152: Deleted "Controlled"*

Response: Deleted as suggested (see Line 158).

**Comment 28:** *Line 154: a bit vague, do you mean the GSD?*

Response: We have replaced bed characteristics with "bed slope and bed roughness" (see Line 160).

**Comment 29:** *Lines 154-155: At the downstream end?*

Response: The plates are embedded at the downstream end of the paved section. We have added this information to the sentence (see Lines 161-162).

**Comment 30:** *Lines 158-159: This reads as if it assumes full familiarity of the previous paper. Revise to be a bit more general*

Response: We have rephrased the sentences on Lines 165-168 to better introduce the term single-grain-size experiments.

**Comment 31:** *Line 162: Report duration?*

Response: We have added some information on the duration of one repetition on Lines 170-171.

**Comment 32:** *Line 165: I am assuming that the j is site specific or for each size class? Can you modify the definition from "mean particle size" to something more specific to make that part more explicit for the reader*

Response: This is a good remark, thanks. At this stage we have indeed not introduced the subscript $j$, which stands for the size class. We have added this information to the definition on Line 174.

**Comment 33:** *Line 167-168: this statement is a bit confusing based on the statement in the last paragraph on L158-159 - I suggest revising this for clarity*

*"paper, we primarily use the single-grain-size experiments conducted in 2018 with the flume configured to match conditions at the Albula field site*

Response: We agree that this was confusing. We have rephrased the entire paragraph (see Lines 176-183).

**Comment 34:** *Line 171: Just for AdN site or across different site set-ups? I find this section to be a bit confusing - would suggest revising for clarity/consistency*

Response: Please refer to our previous answer.

**Comment 35:** *Line 170: Deleted "and"*

Response: We have rephrased this sentence (see Line 181).

**Comment 36:** *Figure 3: Please add word annotations to the figure for each label for ease of reading*

Response: Good suggestion, thanks! We have added annotations (see Line 185).

**Comment 37:** *Lines 176-179: This detail should be in the main text of the paper I think*

Response: We have added this detail on Lines 168-169.

**Comment 38:** *Line 215: Change to comma*

Response: Modified (see Line 226).

**Comment 39:** *Line 218: Deleted space*

Response: Modified (see Line 229).

**Comment 40:** *Line 228: "for the differentiation of multiple"*

Response: Rephrased as suggested (Line 239).

**Comment 41:** *Line 229: "the"*

Response:    Modified (see Line 240).

**Comment 42:** *Line 257: "of"*

Response:    Modified as suggested (see Line 268).

**Comment 43:** *Line 271: "Best separate apparent packets from real packets"*

Response:    Thank you for this suggestion. The sentence has been rephrased (see Line 282).

**Comment 44:** *Line 272: "identified as apparent packings using this criterion"*

Response:    Again, thank you for this suggestion that clarifies the sentence. The sentence has been rephrased accordingly (see Line 283).

**Comment 45:** *Line 284: "allow for the"*

Response:    The word "for" has been inserted and the sentence rephrased (see Lines 294-295).

**Comment 46:** *Line 286: I thought earlier in the paper that the apparent packets introduced bias for the lower threhsolds, which are associated with smaller particles? Do you mean that larger particles generate more apparent packets because they have larger energy and that energy is more likely to show up on the geophones even if the particles arent making direct contact? Can you clarify this*

Response:    You are pointing to the core of the problem. Due to signal attenuation, the apparent packets generated by large impacting particles outside of the plates' boundaries are characterized by small amplitudes, i.e. amplitudes attributed to smaller grain-size classes. This explains the significant scatter of signal responses for the five largest grain-size classes.
We have rephrased this section in order to clarify the origin of this increased scatter visible in Figure 5 (see Lines 297-301).

**Comment 47:** *Line 288: Deleted "see the red boxplots in"*

Response:    Deleted as suggested (see Line 302).

**Comment 48:** *Line 290: Deleted "see the blue boxplots in"*

**Response:** Deleted as suggested (see Line 304).

**Comment 49:** *Figure 4: I suggest adding words to the axes albels for clarity -there are a lot of variables for readers to keep track of in this manuscript and I think it would aid in reading and digestion of the figure*

**Response:** We agree that this could be a good help. In all the following Figures, we have added as many words as possible to the labels, while taking care to not overload the figures too much. Since the variables contained in the labels are also described in the captions, we believe that the current state is a good compromise (see Line 306).

**Comment 50:** *Figure 5: same comment as last figure*

**Response:** Please refer to our previous answer (see Line 311)

**Comment 51:** *Lines 301-303: Is this detail included in the main text of the paper? it seems like it may be useful to make these clear in the text prior to readers encountering this figure*

**Response:** Yes, this information can be found on Lines 292-296.

**Comment 52:** *Line 306: Deleted "calibration"*

**Response:** Deleted as suggested (see Line 320).

**Comment 53:** *Line 306: Deleted comma*

**Response:** Deleted as suggested (see Line 320).

**Comment 54:** *Line 307: allow us to derive the*

**Response:** Modified as suggested (see Line 321).

**Comment 55:** *Line 340: performance of the two calibration methods?*

**Response:** Modified as suggested (see Line 355-356).

**Comment 56:** *Line 345: Deleted "too"*

**Response:** Deleted as suggested (see Line 360).

**Comment 57:** *Line 348: Deleted "as" and ", rather than apparent"*

**Response:** Deleted as suggested (see Lines 363-364).

**Comment 58:** *Line 350: Deleted "as mentioned earlier"*

**Response:** Deleted as suggested (see Line 365).

**Comment 59:** *Figure 7: On c and D it would be helpful to add a second x axis where the grain sizes associated with each of the size classes are delimited in units of length*
*The use of C1-10 for threshold values is different terminology than how these thresholds have been refered to in the main text, I suggest revising the legend to make this consistent*
*Challenging to see the raw data in A due to the overlay, I might suggest puting those boxes behind the data because in this case they are vertically consistent and no important aspects would be obscured*

**Response:** Regarding your first comment on Figure 7, we think that it may be confusing to add the grain size corresponding to the size class. In fact, subplots c and d indicate the number of packets PACK$_j$ located within the class boundaries, and not the size of the particle at the origin of the packet. To clarify, this we have modified the x-axis label.
We have followed your two other suggestions and have modified the figure accordingly (see Line 377).

**Comment 60:** *Line 365: This shielding should be pointed out in the experiment set up more explicitly*

**Response:** We have rephrased the paragraph on Lines 176-183 to set a stronger focus on the shielding.

**Comment 61:** *Line 376: Figures should be introduced in order - modify sentecne to be consistent and have a appear before B*

**Response:** We have decided to keep only one reference to Figure 8 (see Line 391).

**Comment 62:** *Figure 8: Again add words to axes labels*

*The utility of the grey shaded area is unclear to me, it just looks like it is covering the entire region of the plot and ultimately makes all of the envelopes extremely difficult to see*

*It also introduces som ambiguituy in the interpretaiton of the legend. I am assuming that the grey area in the legend is just indicating generically that each colored envelope goes from 5-95th percentile, but then when you look at the plot you can interpret the grey background as that first legend entry. I would suggest modifying the background to be white (which may lead to a change in the erlenbacj color if the issue is the visibility of the yellow)*

*the overlapping envelopes may not be opaque e nough to differentiate, one way to address this could be to also put solid lines on the boundaries of the envelopes to better distinguish them/ highlight the degree of overlap. I would suggest making the median line thicker and adding thin lines on the upper and lower bounds of the envelope for visual clarity*

**Response:** We agree with all your suggestions and have changed Figure 8 accordingly. We have not added words to the y-axis label because, in our opinion, the units given in parenthesis give sufficient information about the meaning of the variable $k_{b,i,j}$ (see Line 401).

**Comment 63:** *Table 4: Include grain size classes explicitly*

**Response:** Changed as suggested (Lines 407-408).

**Comment 64:** *Line 396: "use" or "apply"*

**Response:** We have replaced the word "insert" with "apply" (see Line 411).

**Comment 65:** *Line 397: unit width?*

**Response:** We have changed "the unit fractional flux" to "the fractional flux per unit width" (see Line 412).

**Comment 66:** *Lines 398-399: This sentence is unnecessary, more streamlined to cite figure 9 in previous sentence*

**Response:** As suggested, we have removed this sentence and have added a citation of Figure 9 in the previous sentence.

**Comment 67:** *Lines 400-408: This is a bit tedious as the text just points to details of the figures and tables which will be hard to jump back and forth between once the article is typeset, can you summarize these results more explicitly highlighting important quantitiative valuyes are results directly in the text rather than just saying it is in the figure?*

*I think this would also help streamline things for the reader in what is a fiarly long article.*

**Response:** We agree with you that these two sections were not clearly written and would have hampered a fluent reading. We have rephrased several sentences and have inserted in the text several important values from Table 5 to better underline changes of the accuracy of estimates (see Lines 416-438).

**Comment 68:** *Line 409-418: Same comment as previous paragraph. A sentence or two dedicated to each point on this list with more explicit demonstration of that result/concluision would go a long way in terms of readability*

**Response:** Please refer to our previous comment.

**Comment 69:** *Figure 9: Missing figure caption? Same comments re: labels and grey background as previous figures. Factor 5 is not clear, please modify legend label to make this explicit*

**Response:** Following a comment made by one of the reviewers, Figure 9 had been replaced by a new one using three lines of subplots. We had kept it in the tracked-changes version to facilitate a comparison. We have now removed the old version and have applied your suggestions to the new Figure (see Line 458).

**Comment 70:** *Figure 10: This metric should be introduced in the main text to better prepare the reader to digest this figure. The axes labels should include words for clarity*

*Add text labels to each panel to indicate which represents which method*

*Having now looked at figure 11, I would suggest a revision of the box pltos so readers can more directly compare between the methods*

**Response:** This metric is already described earlier in the text (see Lines 475-476). But we have followed your other suggestions and have added text to the labels and grouped the boxplots into one subplot (see Line 482).

**Comment 71:** *Lines 453-455: Report values to support this interpretation*

Response: We have added the values of the Avançon de Nant site to illustrate the less substantial improvement obtained through the application of the AF method, as opposed to the best improvement observed for the Erlenbach data already mentioned in this paragraph (see Lines 496-498).

**Comment 72:** *Figure 11: Remove grey background - it makes the transparent points and boxes harder to see*

*The way the boxplots are presented to compare methods is not consistent with the previous plot. I would suggest revising towards consistent presentation so the reader is already primed to interpret what they are looking at.*

*Thte use of transparency for the box plots specifically is a bit confusing and makes me feel like I want to ignore the transparent boxes in favor of the opaque ones, by establishing a visual hierarchy that I don't think is necessary. I would suggest perhaps making both boxes opaque but differentiating them in a different way that doesnt place an emphasis on one method as drastically over the other.*

Response: Thanks for these good remarks. We have changed Figure 10 as well as Figure 11 along with their captions to make them consistent and avoid emphasis on one method (see Lines 482 and 501).

**Comment 73:** *Line 482: "was" instead of "were"*

Response: We have kept "were", since we were referring to "the optimal linear coefficient and exponent of the criterion…" (see Lines 522 and 523).

**Comment 74:** *Line 492: Can you provide a quantitative metric to describe the scale of difference? This statement is a bit vague*

Response: Please refer to our answer to your next comment.

**Comment 75:** *Lines 493-494: I think there is a typo in this sentence??*

*What do you mean by "important number of packets"?*

Response: We have rephrased this sentence and have added quantitative information to describe the amount of packets suppressed by the use of AF thresholds (see Lines 534-535)

**Comment 76:** *Lines 498-503: Is this necessary? The result isnt entirely surprising - and I am not sure that this revised method is really introduced anywhere, so in order to streamline the paper, I might suggest removing this*

Response: We agree that the "improvement introduced by the adapted AH method" for the Erlenbach is not surprising. However, the fact that this approach does not improve the signal conversion to fractional transport rates for other sites is also an interesting result that is worth to be reported. Therefore, we decided to keep this paragraph (see Lines 540-545).

**Comment 77:** *Lines 504-508: This paragraph seems to come out of nowhere - especially the mention of the shortcoming of lacking flow velocity measurements. While this may be an important point, I would encourage some revision here to better place this last paragraph into the context of this section*

Response: You are right, this paragraph does not suit very well to this section. We have moved it further down at the end of Section 4.5 (see Lines 650-653).

**Comment 78:** *Figure 12: Same stylistic comments as previous figures*

*The data is so discrete in terms of Vf that it is hard to discern any differences at any individual site as a function of flow velocity*

*I miGHT SUGGEST adding best fit lines to each populdation of data to see if there is any trend in r with v (except for the erlnebach)*

*I would suggest rather than adding random noise to the EB data, just making a box plot - I do not think its appropriate to add random noise here if you don't have measurements because it is likely to misinterpreted by a reader*

Response: In addition to the modifications related to the background color and the text in the labels, we have replaced the data points of the Erlenbach site by a boxplot and have adapted the caption (see Lines 654-659). However, we have decided not to add best fit lines because in our opinion it is already clear enough that there is no obvious trend inside the data. (It might be more helpful to fit a trend line representing all data points, e.g. fitting one to the mean/median values for each velocity class.)